# Induced sensorimotor cortex plasticity remediates chronic treatment-resistant visual neglect

Jacinta O'Shea[1,2,3]*, Patrice Revol[2,4], Helena Cousijn[1], Jamie Near[1†], Pierre Petitet[1], Sophie Jacquin-Courtois[2,5], Heidi Johansen-Berg[1], Gilles Rode[2,5], Yves Rossetti[2,4]

[1]Wellcome Centre for Integrative Neuroimaging, Oxford Centre for Functional MRI of the Brain, Nuffield Department of Clinical Neurosciences, University of Oxford, Oxford, United Kingdom; [2]Lyon Neuroscience Research Center, ImpAct (Integrative, Multisensory, Perception, Action & Cognition) team INSERM U1028, CNRS UMR5292, University Lyon 1, Bron, France; [3]Donders Institute for Brain, Cognition and Behavior, Radboud University, Nijmegen, Netherlands; [4]Hospices Civils de Lyon, Mouvement et Handicap, Hôpital Henry Gabrielle, Saint Genis-Laval, France; [5]Hospices Civils de Lyon, Service de Rééducation Neurologique, Hôpital Henry Gabrielle, Saint Genis-Laval, France

*For correspondence:
jacinta.oshea@ndcn.ox.ac.uk

Present address: †Douglas Mental Health University Institute and Department of Psychiatry, McGill University, Québec, Canada

**Abstract** Right brain injury causes visual neglect - lost awareness of left space. During prism adaptation therapy, patients adapt to a rightward optical shift by recalibrating right arm movements leftward. This can improve left neglect, but the benefit of a single session is transient (~1 day). Here we show that tonic disinhibition of left motor cortex during prism adaptation enhances consolidation, stabilizing both sensorimotor and cognitive prism after-effects. In three longitudinal patient case series, just 20 min of combined stimulation/adaptation caused persistent cognitive after-effects (neglect improvement) that lasted throughout follow-up (18–46 days). Moreover, adaptation without stimulation was ineffective. Thus stimulation reversed treatment resistance in chronic visual neglect. These findings challenge consensus that because the left hemisphere in neglect is pathologically over-excited it ought to be suppressed. Excitation of left sensorimotor circuits, during an adaptive cognitive state, can unmask latent plastic potential that durably improves resistant visual attention deficits after brain injury.
DOI: https://doi.org/10.7554/eLife.26602.001

## Introduction

Stroke is a leading cause of adult disability (*Adamson et al., 2004*). The majority of right hemisphere stroke survivors suffer acute 'neglect' – lost awareness of left space (*Figure 1A–C*) (*Buxbaum et al., 2004*; *Stone et al., 1993*). While endogenous plasticity drives neglect recovery during the sub-acute stage (≤3 months post-stroke), recovery plateaus in the chronic stage (*Ramsey et al., 2016*). Chronic neglect predicts poor functional outcome, entailing prolonged hospitalization, reduced independence, and lasting disability (*Jehkonen et al., 2006*). There is currently no clinically established effective treatment (*Bowen et al., 2013*). Several single-session experimental interventions can improve neglect transiently (minutes to hours), with intensive repetition over days/weeks/months prolonging gains, but this is costly, and dose-response data are lacking (*Kerkhoff and Schenk, 2012*). Here we present initial proof-of-concept scientific evidence for the efficacy of a novel experimental intervention. This single-session protocol used non-invasive brain stimulation to enhance consolidation of behavioural therapy, resulting in long-lasting improvements in visual neglect.

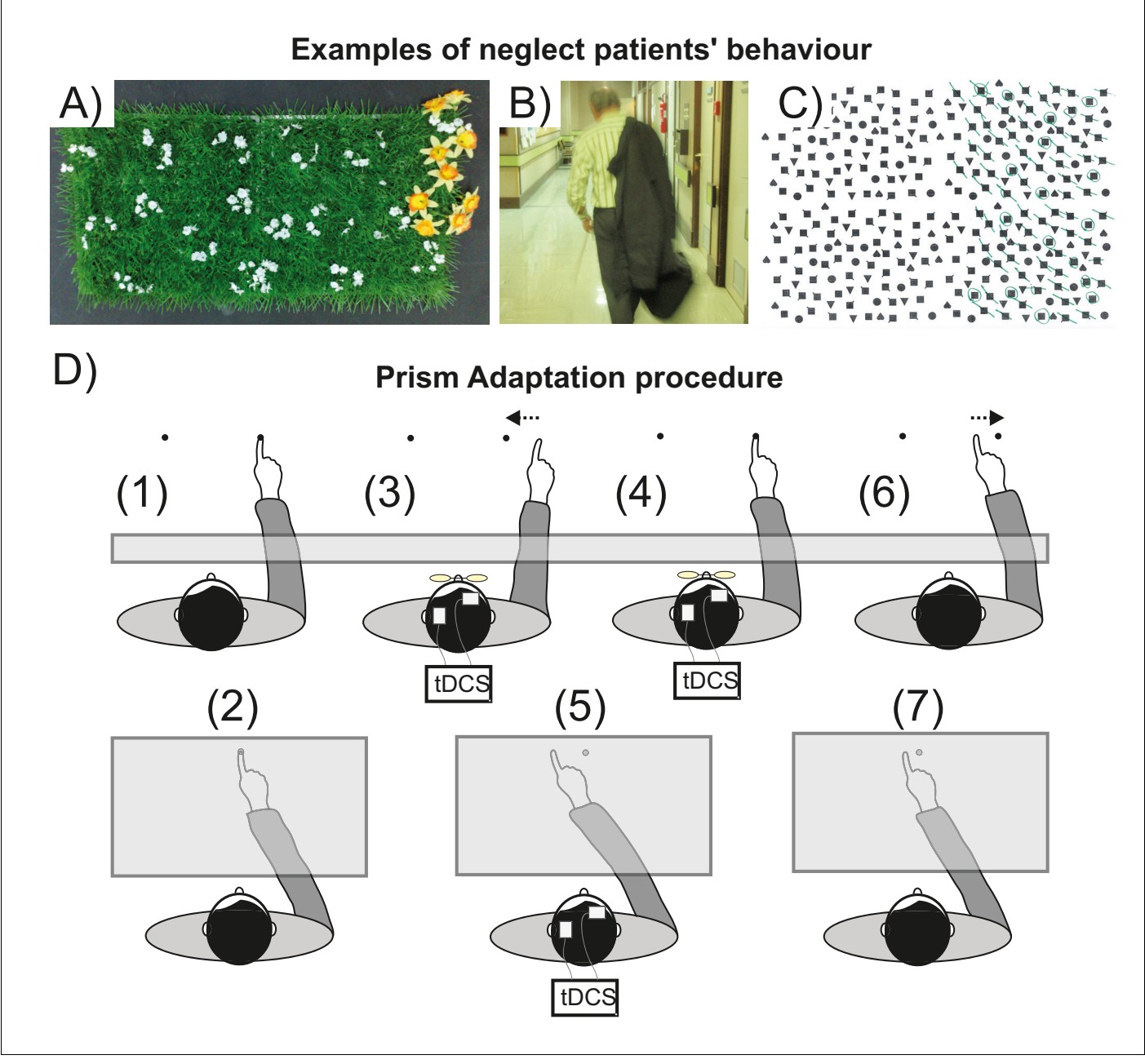

**Figure 1.** Visuospatial neglect and prism adaptation. (A–C) Examples of neglect behavior by patients in the present study. (A) Gardening task: arrange the flowers uniformly around the garden; (B) patient has neglected the left of his body; (C) Ota-Girardi task: cancel all targets on the page. (D) Prism Adaptation. (1) Baseline: visual feedback. Participants make rapid centre-out pointing movements to targets located at 10 degrees left or right. Vision of the hand start position is occluded. (2) Baseline: no visual feedback. Participants point at a central target. Vision of the hand is occluded throughout. (3) Adaptation: early prism exposure. Participants repeat (1) wearing 10° right-shifting prisms and initially make large rightward pointing errors. They use error feedback to correct their movements from right to left. (4) Adaptation: late prism exposure. After sustained prism exposure participants successfully realign hand-eye coordination leftward to regain baseline accuracy (i.e. they adapt). (5) Adaptation: prism after-effect. As participants adapt, this induces a leftward bias (prism after-effect), measured as the change from baseline (2). (6) Washout. After prism removal when participants point as in (1), the effect of adaptation is seen as a leftward error that is quickly corrected to restore baseline accuracy. (7) Retention. The magnitude of any remaining prism after-effect is measured as in (2).

DOI: https://doi.org/10.7554/eLife.26602.002

Prism adaptation (PA, *Figure 1D*) is a form of behavioural therapy in which neglect patients gradually realign hand-eye coordination leftward to adapt to a rightward optical shift (*Rossetti et al., 1998*; *Serino et al., 2009*). A typical adaptation session lasts ~20 min, and induces a leftward pointing bias that persists for several minutes after prism removal - the prism 'after-effect' (AE). In neglect, but not in healthy individuals, this acquired leftward bias may also generalize abnormally, transferring to untrained cognitive domains (eg: visual search, reading, wheelchair driving), thus improving neglect (*Newport and Schenk, 2012*). Here we address two key problems that limit PA's potential future transformation from an experimental intervention into an established clinical treatment (*Barrett et al., 2012*): (1) not all patients respond therapeutically; and (2) the benefit of a single session is transient (~1 day).

In this scientific proof-of-concept study, we addressed both problems in a series of longitudinal sham-controlled case studies in three patients with chronic, treatment-resistant neglect. We targeted anodal transcranial direct current stimulation (a-tDCS) stimulation at left primary motor cortex (M1) while patients adapted to prisms with the contralateral right hand.

We targeted M1 based on evidence that this brain region mediates the early consolidation of motor memories formed during adaptation (*Hadipour-Niktarash et al., 2007*; *Hunter et al., 2009*; *Landi et al., 2011*; *Li et al., 2001*; *Richardson et al., 2006*).

We chose tonic excitatory stimulation (a-tDCS) because of evidence that this intervention changes motor cortex physiology in a manner that promotes neural and functional plasticity. M1 a-tDCS increases motor cortico-spinal excitability (*Nitsche and Paulus, 2001*; *Nitsche et al., 2005*), lowers motor cortical inhibitory tone (*Stagg et al., 2009*), and strengthens synaptic efficacy of connections that are intrinsically active (*Fritsch et al., 2010*). These physiological mechanisms, interacting with endogenous task-related brain activity, likely mediate the enhancement effects of M1 a-tDCS on motor learning and memory formation that can be observed when stimulation is applied during a range of motor tasks (*Buch et al., 2017*; *Galea et al., 2011*; *Hunter et al., 2009*; *Panico et al., 2017*; *Reis et al., 2008*).

We reasoned that applying plasticity-promoting stimulation (a-tDCS) to M1, while participants are in a task state (adaptation) that engages the targeted brain region, would enhance specifically that function (early consolidation) for which M1 is specialized. Hence, by tonically disinhibiting M1 via a-tDCS during adaptation, we aimed to potentiate the earliest memory traces formed during adaptation. We predicted that M1 a-tDCS would thus enhance consolidation of adaptation, stabilizing consequent prism after-effects in both the sensorimotor (leftward pointing bias) and cognitive (neglect improvement) domains.

To test this hypothesis, we used a two-step bench-to-bedside translational approach. First, we demonstrated that M1 a-tDCS during PA stabilized the pointing AE in healthy volunteers and in a single case neglect patient. Next we assessed consequent cognitive gains longitudinally in case studies of three patients with chronic, severe, treatment-unresponsive neglect. All showed no response to PA therapy alone. However, after PA combined with M1 a-tDCS, all patients exhibited long-lasting improvements in neglect. Remarkably, the clinical benefit after a single 20 min intervention persisted throughout weeks of follow-up with no return to baseline. Thus, excitatory stimulation of M1 synergized PA, potentiating the prism after-effect, and inducing lasting, stable gains in chronic, resistant visual neglect.

## Results

### Experimental phase

Healthy volunteers adapted to 10° right-shifting prisms by making rapid pointing movements trial-by-trial with the right hand to either of two dot targets located at 10° left/right on a table in the reach space in front of them (*Figure 1D*). During prism exposure, visual, proprioceptive, and motor feedback of the rightward pointing errors (induced by the optical shift) enabled participants to gradually correct their errors from right to left over time to re-gain baseline accuracy. The consequent AE was measured as a leftward error (change relative to baseline) when participants pointed without prisms at a central, untrained target. On all AE trials, visual feedback of the target and hand position was occluded, so participants had to rely solely on proprioceptive and motor feedback signals to guide reach accuracy (*O'Shea et al., 2014b*).

During the Adaptation phase, participants underwent 100 trials of prism exposure spaced across blocks, interleaved with blocks of AE measurement (*Figure 2A*). This design encouraged gradual adaptation, which itself favours retention (*Michel et al., 2007*), and, critically, allowed for tDCS (tonic stimulation) to be applied continuously throughout the Adaptation phase (20 min). *Figure 2A* shows the resulting learning and memory dynamics: participants corrected their errors rapidly within prism exposure blocks (E1-6), but decay occurred between blocks. The magnitude and stability of the consequent AE also evolved across blocks (AE1-6). Since the stimulation goal was to enhance AE consolidation, tDCS (real or sham) was applied throughout the Adaptation phase. The critical test was the persistence of the AE memory trace, which was assessed by measuring the magnitude of AE decay in the subsequent Washout and Retention phases. Across the Adaptation phase, the AE magnitude stabilized progressively (*Figure 2A*). We next assessed AE decay during active washout, an interference measure of the strength of the AE memory trace. During Washout prisms were removed and participants again made rapid pointing movements to the left and right targets. As a consequence of adaptation, they initially made leftward errors and used feedback to correct their errors rightward to regain baseline accuracy. Consequently, the AE decayed toward baseline over six interleaved blocks. Retention of the AE was assessed after a 10 min rest delay.

Statistical analyses using repeated measures ANOVA and 1-tailed planned contrasts (anodal >sham) tested the *a priori* 1-tailed directional hypothesis that M1 a-tDCS during Adaptation would enhance AE consolidation. This key prediction was confirmed: after M1 a-tDCS (but not sham control tDCS) the AE persisted throughout Washout and Retention (*Figure 2A,B*).

*Figure 2* summarizes the AE results for this key experiment and for several control experiments. All experiments used repeated measures designs and order counterbalancing. Experiment 1 tested the key hypothesis that M1 a-tDCS enhances consolidation when applied during Adaptation (*Figure 2A,B*). Experiment 2 tested the hypothesis that this would not occur with the same stimulation applied at rest prior to Adaptation (*Figure 2C*). To determine the anatomical specificity of the M1 stimulation effect (shown in *Figure 2A,B*), control experiments 3 and 4 applied the same stimulation during Adaptation, but to left parietal cortex (*Figure 2D*) and right cerebellum (*Figure 2E*), since these two regions form part of the sensorimotor circuit that controls adaptation with the right hand. Lesions to parietal cortex (*Newport et al., 2006*; *Pisella et al., 2004*) and cerebellum (*Martin et al., 1996*; *Weiner et al., 1983*) have been shown to disrupt the magnitude of error correction and prism after-effects, but have not been shown to specifically affect consolidation.

Statistical tests (full details in Appendix 2, *Supplementary file 1*) confirmed the following findings: (1) M1 a-tDCS increased AE magnitude during Adaptation (*Figure 2B*, Anodal-Sham p=0.042, d = 0.5). (2) M1 a-tDCS during Adaptation enhanced subsequent AE persistence during Washout and Retention (*Figure 2B*) (Anodal-Sham: washout p=0.008, d = 0.944; retention p=0.034, d = 0.911). This is the key effect. (3) This enhanced AE persistence was a specific consequence of excitatory (not inhibitory) M1 stimulation (*Figure 2B*) (Cathodal-Sham: washout p>0.4; retention p>0.8). (4) Enhanced AE persistence occurred with M1 a-tDCS during but not before PA (*Figure 2B* versus C) (interaction: cognitive state ×tDCS: washout p=0.004; retention p=0.019). (5) a-tDCS during PA applied to left posterior parietal cortex (17) (*Figure 2D*) or right cerebellum (18) (*Figure 2E*) had no such effect, demonstrating anatomical specificity to M1 (Anodal-Sham Region x tDCS interaction: PPC versus M1: washout p=0.025, retention p=0.044; CB versus M1 washout p=0.027, retention p=0.033). (6) M1 a-tDCS had no effect on the error correction rate, neither during Adaptation nor Washout (*Figure 2A*; all p>0.9). Also: (7) A repeat of the protocol in *Figure 2A* (*Figure 2—figure supplement 1*) using sham prisms confirmed that neither pointing alone nor pointing during M1 a-tDCS induced a leftward bias, confirming that our key finding (*Figure 2A*) reflects a synergistic effect of stimulation on PA, and not an effect of a-tDCS alone.

In summary, the predicted effect was highly specific: excitatory but not inhibitory stimulation (*Figure 2B*), applied to left M1, but not left parietal (*Figure 2D*) or right cerebellar cortex (*Figure 2E*), during but not prior to adaptation (*Figure 2B,C*), enhanced retention of the leftward pointing bias caused by real (but not sham) prism adaptation (*Figure 2A*, *Figure 2—figure supplement 1*). This effect was observed only when visual feedback was occluded (*Figure 2A*, AE7-15). Stimulation had no effect on pointing accuracy when visual feedback was present - not during prism exposure (*Figures 2A*, E1–6), washout (W1-6), or during pointing with sham prisms (*Figure 2—figure supplement 1*).

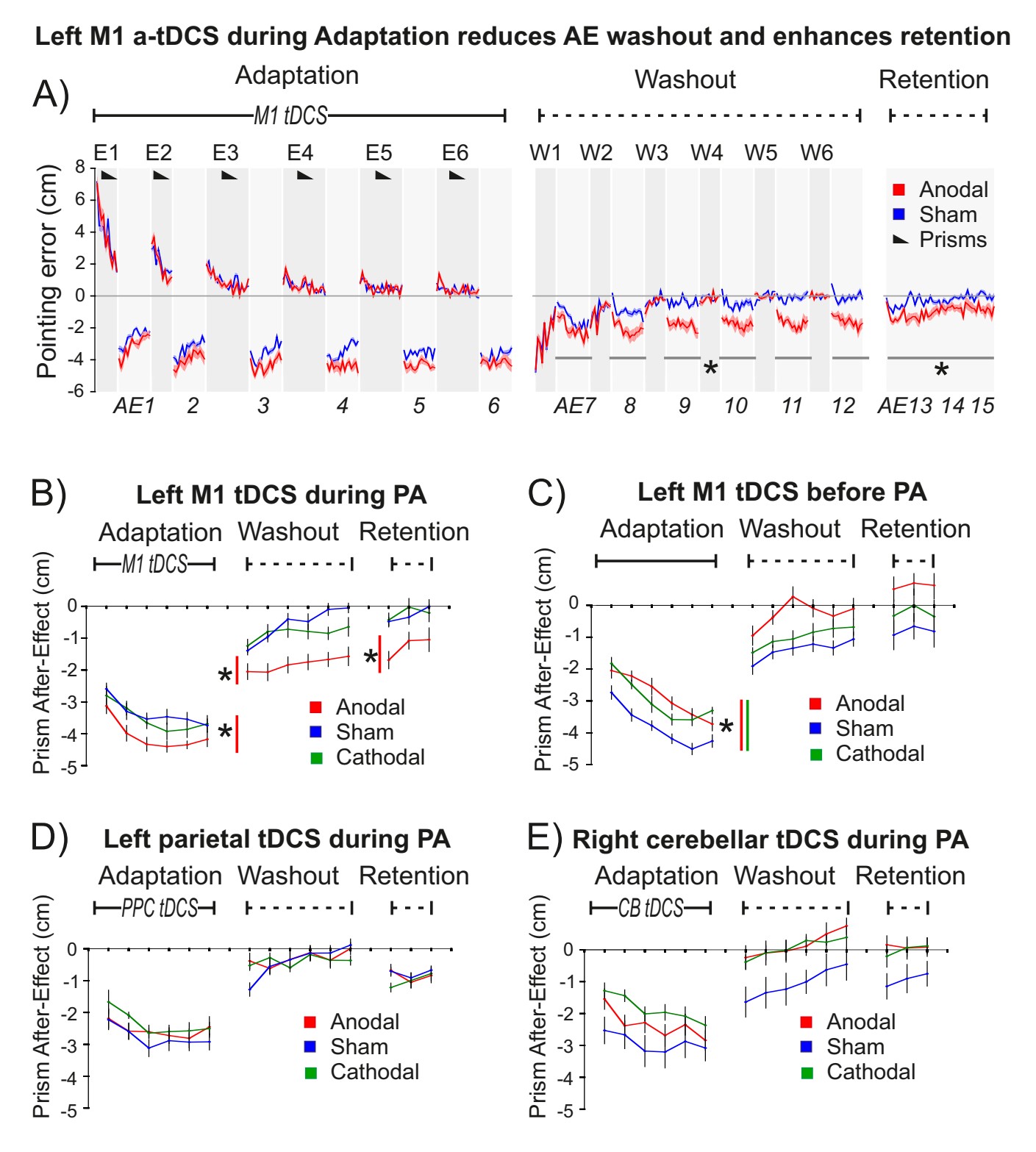

**Figure 2.** M1 a-tDCS during Prism Adaptation (PA) enhanced consolidation of the prism after-effect. The x-axis represents baseline accuracy (zero error), with prism after-effect (AE) data representing changes from baseline. For each panel N = 9. Asterisk indicates significant difference between Anodal and Sham (p<0.05). (A) Pointing accuracy in healthy volunteers when anodal (red) or sham (blue) tDCS was applied to M1 during Adaptation (Experiment 1). Black wedges indicate blocks throughout which prisms were worn. During Adaptation (prism exposure, (E1–E6) and Washout (prisms

*Figure 2 continued on next page*

*Figure 2 continued*

removed, (W1–W6), participants saw the outcome of the trial, so could correct their errors. The AE was measured without visual feedback (AE1-15, shaded light grey). Solid lines show pointing accuracy averaged across participants (shading =±1 SEM). Adaptation and Washout lasted 20 min each. Retention lasted 6 min after 10 min of blindfolded rest. Relative to sham, anodal tDCS increased AE persistence throughout Washout and Retention: note the leftward shift in AE7-15 (no visual feedback), whereas accuracy in interleaved blocks W1-6 is indistinguishable. (B–E) Prism after-effect in different stimulation conditions. Panel (B) summarizes group mean AEs (±1 SEM) for the dataset shown in (A) (AE1-15 only) and also shows data for reversed polarity stimulation (cathodal, green). Other panels plot the same summary AE data for stimulation of: (C) M1 before Adaptation (Experiment 2); (D) right posterior parietal cortex (PPC) during Adaptation (Experiment 3); E) right cerebellum (CB) during Adaptation (Experiment 4).

DOI: https://doi.org/10.7554/eLife.26602.003

The following figure supplement is available for figure 2:

**Figure supplement 1.** Control experiment 5: M1 a-tDCS alone does not cause a leftward shift in pointing behavior.

DOI: https://doi.org/10.7554/eLife.26602.004

*Figure 3* (Experiment 6) shows a repeat of the experiment in *Figure 2A*, but with no Washout phase and with retention measured over 5 days. After Adaptation, although AE magnitudes decayed over time (p<0.001), this was slowed by M1 tDCS (Anodal-Sham: p=0.002, d = 1.263; no tDCS ×Time interaction), resulting in significant AE persistence across days after real but not sham stimulation. Relative to participants' mean pointing accuracy at baseline (over 4 days prior to PA), pointing remained significantly left-shifted in the 4 days after PA specifically in the anodal tDCS condition (*Figure 3—figure supplement 1*) (anodal-baseline p=0.001, d = 0.73; sham-baseline p>0.5). Hence, the stimulation-enhanced AE persisted across days while participants simply carried out their

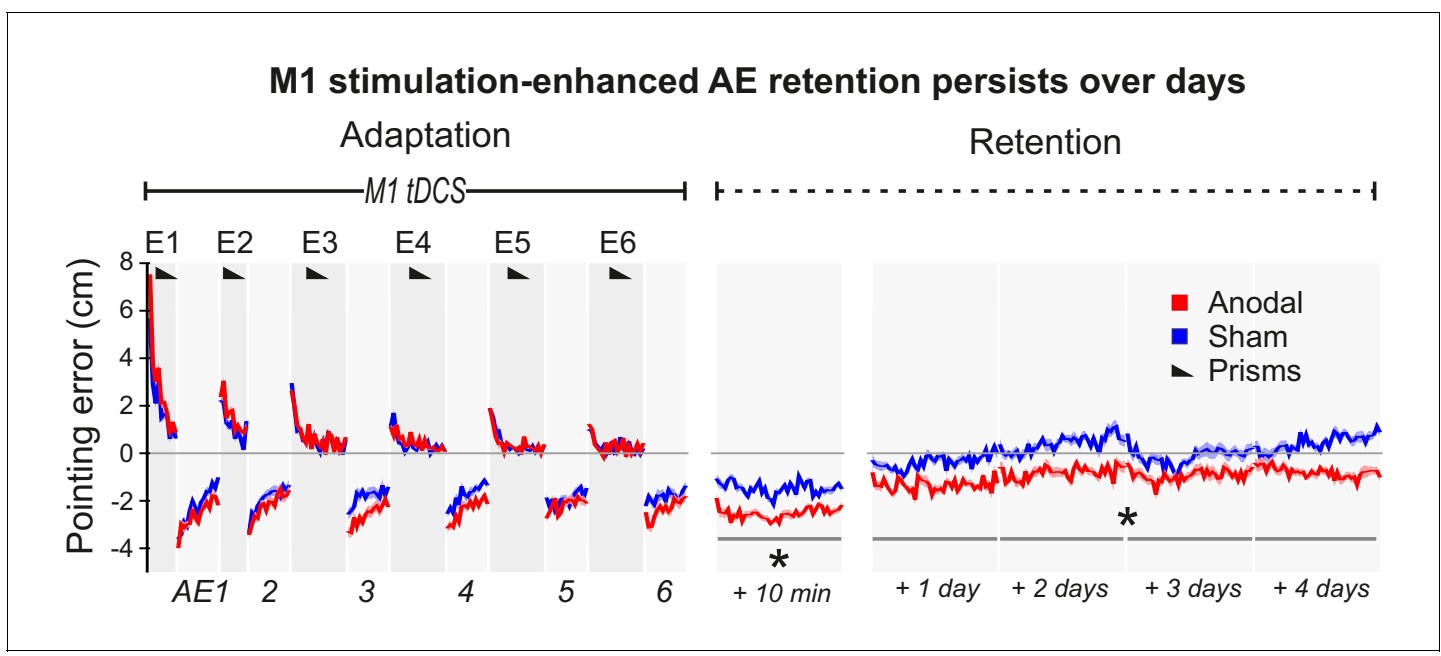

**Figure 3.** M1 a-tDCS during Adaptation enhanced prism after-effect retention across days. Pointing errors in healthy volunteers (N = 10) when anodal (red) or sham (blue) stimulation was applied during Adaptation (Experiment 6). Compared to the experiment of *Figure 2A*, there was no active washout phase. Instead, retention was measured daily over 5 days after adaptation. Solid lines show pointing accuracy averaged across participants (±1 SEM, shading). As in the experiment shown in *Figure 2A*, the AE persisted throughout the measured retention period only when M1 a-tDCS had been applied during Adaptation. Asterisks indicate significantly larger AEs in the anodal versus sham condition throughout Retention (p<0.05).

DOI: https://doi.org/10.7554/eLife.26602.005

The following figure supplement is available for figure 3:

**Figure supplement 1.** M1 stimulation-enhanced retention persists across days.

DOI: https://doi.org/10.7554/eLife.26602.006

normal daily activities, indicating a robust memory trace that endured over a timescale of potential clinical relevance despite multiple changes of context.

By what causal mechanism did M1 stimulation enhance AE consolidation? We hypothesized that a known effect of M1 a-tDCS, transient reduction of M1 cortical inhibitory tone, measured as a reduction in gamma-aminobutyric acid (GABA) concentration (*Stagg et al., 2009*), may partly mediate the behavioural effect. In other words, we reasoned that differences in individuals' physiological sensitivity to M1 a-tDCS should causally constrain the magnitude of behavioural change that such stimulation could induce. Specifically, we predicted that, across individuals, the magnitude of M1 a-tDCS-induced physiological change (i.e. expected M1 GABA decrease) would co-vary with the magnitude of stimulation-induced behaviour change (i.e. expected AE retention increase). In the absence of this physiological response to stimulation, the predicted behavioural consequence should not occur. To test this, in experiment 7 we repeated the experiment in *Figure 3* in a new cohort, and quantified for each individual the relative change (anodal-sham) in percent AE retention 1 day after PA was combined with sham versus M1 a-tDCS. To quantify GABA change, participants underwent a magnetic resonance spectroscopy brain scan before and after

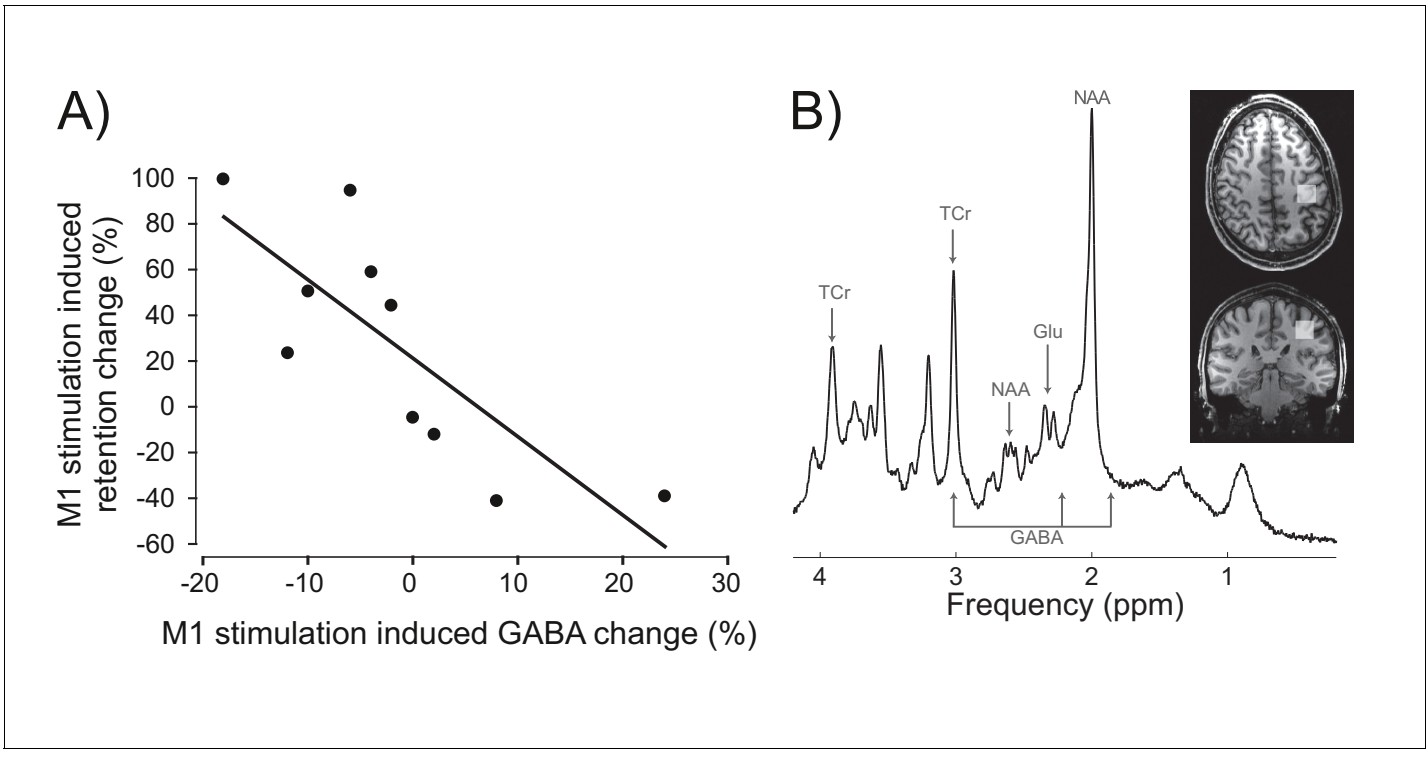

**Figure 4.** Inter-individual differences in neurochemical response to stimulation (M1 GABA change) co-varied with changes in prism after-effect retention caused by M1 a-tDCS. (A) Experiment 7: X-axis shows individuals' percent change in GABA concentration after M1 a-tDCS (post-pre), quantified as ratios of total creatine. Y-axis shows individuals' percent change in AE retention 24 hr after PA + tDCS (anodal-sham). Correlation (N = 10) shows quantitative covariation between the behavioural and neurochemical effects of M1 a-tDCS. Note the data pass through the origin. (B) Representative image for a single participant showing voxel placement in the hand knob region of M1 and spectral quality. Frequency spectrum labels (ppm = parts per million) indicate peaks for GABA, Glutamate (Glu), total Creatine (TCr = creatine + phosphocreatine) and N-acetylaspartate (NAA).
DOI: https://doi.org/10.7554/eLife.26602.007

The following source data and figure supplements are available for figure 4:

**Source data 1.** Individual participants' magnetic resonance spectroscopy data for M1 and occipital cortex before and after stimulation.
DOI: https://doi.org/10.7554/eLife.26602.010
**Figure supplement 1.** AE retention scores (%) 24 hr after PA was combined with anodal or sham tDCS in Experiment 7.
DOI: https://doi.org/10.7554/eLife.26602.008
**Figure supplement 2.** MRS data from occipital cortex voxel in Experiment 7.
DOI: https://doi.org/10.7554/eLife.26602.009

M1 a-tDCS was applied at rest. Correlation tests confirmed our *a priori* one-way directional prediction of a significant positive association between the neurochemical and behavioural effects of stimulation (*Figure 4*) ($r_s$ (10)=−0.83, p=0.0015, 1-tail). Across individuals, when GABA decreased in response to tDCS, relative (anodal-sham) AE retention increased. Individuals who did not show this expected physiological response to excitatory anodal tDCS (i.e. disinhibition, GABA decrease) also did not show behavioural enhancement (AE retention increase). If GABA did not change, retention did not change. If GABA increased post-stimulation, AE retention decreased. In sum, this pattern of quantitative covariation between the neurochemical and behavioural effects suggests that M1 a-tDCS-enhanced AE consolidation depends causally, at least in part, on a reduction of M1 cortical inhibitory tone. Of course it cannot be ruled out that some unknown third variable influences both the tDCS-induced GABA change (at rest) and the tDCS-induced behaviour change (during adaptation), thus accounting for their covariation. Control analyses (see SI) showed that this quantitative pattern of neurochemical-behavioural covariation was neurochemically specific (GABA not glutamate) and anatomically specific (M1 not occipital cortex) (all p>0.05). Full details of behavior, metabolites and analyses are in SI (see also *Figure 4—figure supplements 1,2*, *Figure 4—source data 1*).

## Clinical phase

We next tested the clinical impact of M1 a-tDCS during PA in three patients with chronic, treatment-unresponsive visual neglect. All three patients had shown clinical improvements after PA therapy at the sub-acute stage, but at the time of testing were no longer responsive to PA therapy.

To first confirm that the stimulation-enhanced pointing AE retention demonstrated above also occurs in neglect, Patient 1 experienced the procedure illustrated in *Figure 2* (Adaptation, Washout, Retention) on four occasions over 4 months. Only on the third occasion was true anodal stimulation applied, the other three sessions used sham (S-S-A-S). 95% confidence intervals were

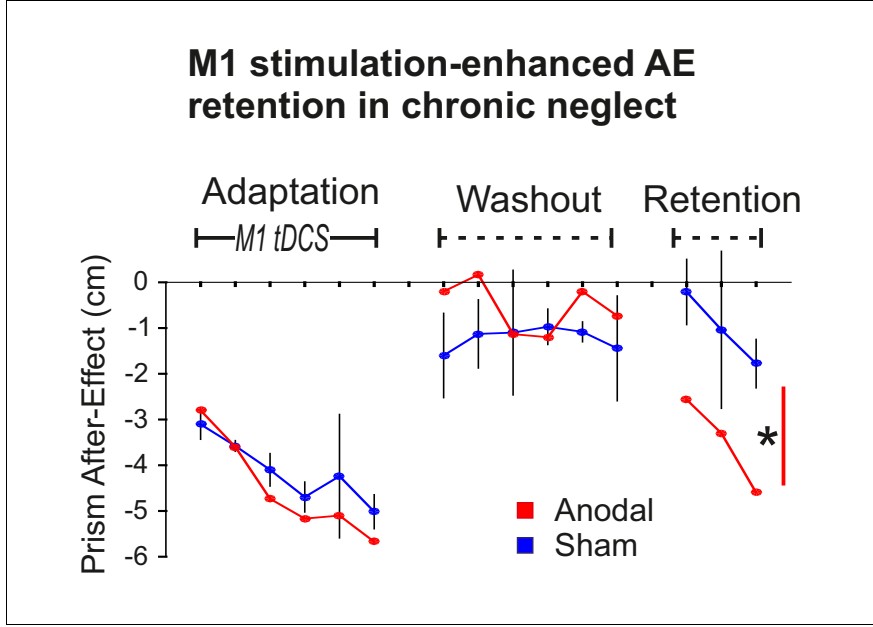

**Figure 5.** Stimulation-enhanced retention in chronic neglect. The prism after-effect over time for anodal (red) and sham (blue) stimulation of M1 during Adaptation in a single case neglect patient (Patient 1). Data are displayed as described for *Figure 2B* except that sham data are the mean (±95% confidence intervals) across 3 sessions of the protocol, and anodal data are from a single session. AE retention was significantly enhanced by stimulation in this neglect patient, similar to the healthy volunteers shown in *Figure 2A,B*. Asterisk indicates significantly greater retention for the planned contrast of anodal versus sham in Retention blocks 1 and 3 (*p<0.001).
DOI: https://doi.org/10.7554/eLife.26602.011

**Figure 6.** Patients' Lesion Anatomy. (A–C) T1-weighted structural scans of Patients 1–3, respectively. (D) Grey matter lesion overlap between the three patients in group mean MNI space. (E) 'Disconnectome map' shows white matter damage overlap between the three patients. Red indicates white matter damage overlap in all three patients in the right inferior fronto-occipital fasciculus (rIFOF), the right superior longitudinal fasciculus (rSLF) and the right anterior thalamic radiation (rATR).

DOI: https://doi.org/10.7554/eLife.26602.012

computed on the patient's typical behaviour (the three sham sessions). *Figure 5* shows that AE magnitudes in the anodal condition were largely within the normal range during Adaptation and Washout, but diverged during Retention. During Adaptation, AE magnitudes increased progressively, as a function of interleaved blocks of prism exposure (not shown). During Washout, a small AE persisted in both conditions. After a 10 min rest delay, this AE, which had decayed during Washout, re-emerged towards the end of Retention, an adaptation memory rebound phenomenon known as spontaneous recovery (*Kording et al., 2007*). Statistical analysis confirmed a significant AE was retained in both conditions. The retained AE was larger for anodal versus sham tDCS (p<0.001, d = 2.382). Hence, the enhanced AE retention effect that was predicted and observed in healthy participants (*Figure 2B*) was also present in this neglect patient.

To test the key hypothesis that M1 a-tDCS would enhance the efficacy of PA therapy, three patients (Patient 1 and 2 others) with chronic, treatment-unresponsive neglect (*Figure 6*) underwent serial longitudinal neuropsychological assessments before and after PA combined with sham or anodal tDCS of M1. Statistical analyses tested for improvements in neglect after versus before PA + tDCS (RM ANOVA: Time, tDCS; planned contrasts: post-pre, anodal-sham; full statistics in SI). In summary, after each PA +M1 a-tDCS intervention, every patient showed a significant lasting improvement in neglect. By contrast, there was no response to PA + sham.

Patient 1 was assessed over 211 days, in two phases, in each of which the effect of PA + M1 a-tDCS was contrasted with PA + sham. In phase A, during the 2–3 weeks before and after each of the last 2 of the 4 PA sessions described in *Figure 5* (ie: S-S-A-S), Patient 1 completed a battery of 6 neuropsychological tests of neglect (SI: Methods) (*Figure 7A*). After PA + a-tDCS, Patient 1's neglect scores improved by ~20% absolute, ~50% relative (*Figure 7A*) (post-pre: p=0.005, d = 0.578). This gain was still present 66 days later, at the last test before the PA + sham session. By contrast, there was no further change in neglect after the subsequent PA + sham intervention (p>0.9). Rather, the gains observed after the PA + anodal stimulation session were maintained for at least a further 22 days (i.e. 88 days in total). Statistics confirmed a significantly greater improvement in neglect after PA +a tDCS compared to PA +sham (p=0.011, d = 1.52).

Phase B (77 days) in Patient 1 repeated phase A, but with a PA procedure that included Adaptation only (no Washout), with the order of sham and anodal interventions reversed, and with a new battery of 10 neglect tests (SI: Methods). By changing the neglect tests, we aimed to minimize potential learning effects, and by reversing the order of stimulation (S-A) we addressed the possibility that a ceiling effect had precluded detecting further clinical improvement in the Phase A sham condition. In Phase B Patient 1 scored ~50% at baseline (*Figure 7B*), and just as in Phase A, there was no change in neglect after PA + sham stimulation (p>0.37). By contrast, there was once again a significant and long-lasting improvement in neglect after PA + M1 a-tDCS (20% absolute, 40% relative)(post-pre: p<0.001, d = 0.773). This gain was still present 46 days after the intervention, the latest time-point tested. Statistics confirmed a significantly greater improvement in neglect after PA + a-tDCS compared to PA +sham (p<0.001, d = 1.697). Analysis of all Patient 1's data across the phase reversal design (i.e. A-S-S-A) further confirmed this key prediction (main effect tDCS: p=0.001).

Phase B (only) was repeated in two further treatment-unresponsive neglect patients (*Figure 7C–D*; *Figure 7—figure supplement 1*). Patient 2 received stimulation in the order anodal-sham, Patient 3 sham-anodal. Both showed no improvement after PA + sham stimulation, but significant improvements after PA + anodal stimulation. Patient 2 showed a large effect (30% absolute, 53% relative; p=0.003, d = 1.836), while patient 3 showed a small effect (7% absolute, 14% relative; p=0.05, d = 0.286). The direct contrast of anodal versus sham absolute percent change in neglect was significant for Patient 2 (p=0.002, d = 0.714) and trend level for Patient 3 (p=0.107, d = 0.434).

Group statistics combining all three patients on Phase B absolute percent change data (*Figure 7C*, *Figure 7—source data 1*) confirmed the key prediction (Anodal >Sham: p=0.005, d = 0.899). *Figure 7C* shows the improvement in neglect in Phase B averaged across all three patients, and *Figure 7D* shows the individual patients' results at different times after PA + M1 a-tDCS. In all four anodal > sham case study comparisons (i.e. Phase A in Patient 1, Phase B in Patients 1–3), patients showed no response to PA therapy plus sham, but when coupled with M1 a-tDCS, PA caused lasting clinical gains in every case. Note that stimulation order was counterbalanced across the aggregate of these four studies (Patient 1: A-S and S-A; Patient 2: A-S; Patient 3:

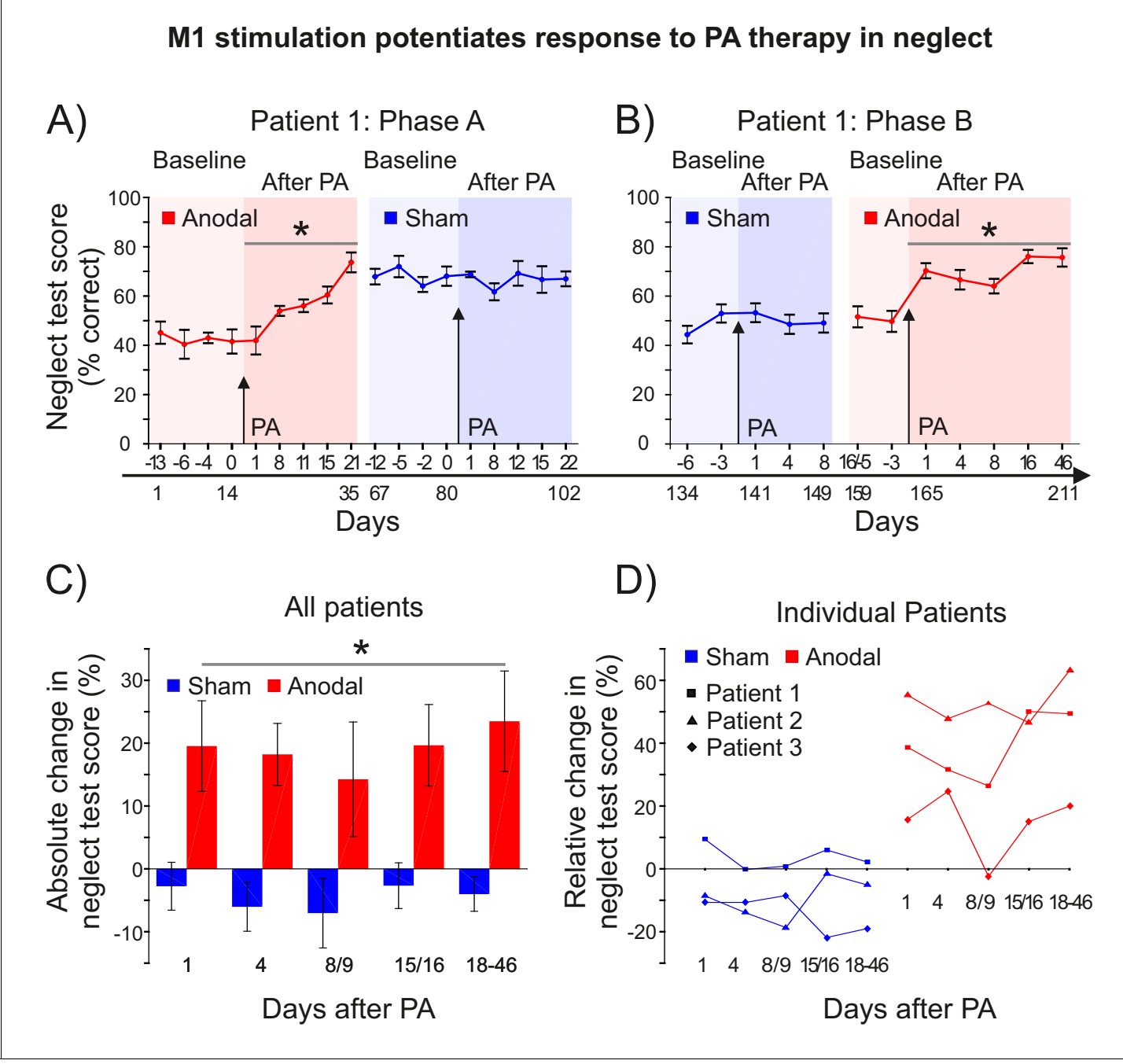

**Figure 7.** M1 stimulation during PA leads to lasting improvements in neglect. (**A**) Behavioural measures of neglect in a single case chronic neglect patient (Patient 1) before and after PA sessions combining M1 anodal (red) and sham (blue) stimulation during adaptation. Datapoints are mean % neglect score ±1 SEM. A score of ~50% indicates the patient completed the rightmost half of the six tests in the battery. (**B**) As (**A**), but with a different battery of 10 neglect test measures, reversal of the order of anodal and sham stimulation, and no Washout phase during prism adaptation. (**C, D**) Percentage changes in performance on a battery of neglect tests over time after PA +M1 a-tDCS, averaged across all three patients (±1 SEM) (**C**), and for each individual case (**D**). Asterisks indicate significant improvement in neglect score after PA +M1 a-tDCS compared to sham (p<0.05).
DOI: https://doi.org/10.7554/eLife.26602.013

The following source data and figure supplement are available for figure 7:

**Source data 1.** Individual patients' percent correct scores on neglect batteries before and after PA + tDCS at each timepoint and test phase.
DOI: https://doi.org/10.7554/eLife.26602.015

**Figure supplement 1.** M1 stimulation during PA leads to lasting improvements in neglect.
DOI: https://doi.org/10.7554/eLife.26602.014

S-A), which suggests that the null response to PA + sham tDCS in each case was not simply a ceiling effect, and that the gains from PA + real stimulation cannot be simply explained as generalized learning of the neglect tests.

Given that PA + M1 a-tDCS induced a sustained leftward shift, both in sensorimotor (pointing) and spatial cognition (neglect) prism after-effects, we addressed the possibility of unintended leftward reach impairments (i.e. ataxic side effects) in patients. We recorded detailed kinematic measures of the right arm while patients performed a visually-guided naturalistic spatial reaching task, once each in the days before and after PA (+ tDCS). For detailed methods, data and analyses see Appendix 2 (*Supplementary file 2*). On each trial, the task required patients to perform a centre-out reaching movement with the ipsilesional (right) arm to grasp a ball, and then place it in a basket in the left or right hemifield, depending on a prior central visual instruction cue. Only the centre-out reach phase was analysed. Hence, the movement was identical on each trial, except that a central pre-cue at reach onset instructed the patient that they would have to perform either a leftward or rightward reach next, after they had grasped the ball. Movements were self-paced (i.e. no speed instructions). Since neglect patients commonly exhibit "directional hypokinesia" - slowing of contralesionally directed (i.e. leftward) movements - we expected the spatial cueing element of this task to make it sensitive to neglect. Statistical analysis confirmed this: at baseline, despite the identical centre-out reaching movement required on every trial, patients exhibited slower movement times when pre-cued that they would have to subsequently place the ball in the left hemifield compared to the right (Left >Right, p=0.005). This hemifield difference (L > R) was abolished by prism adaptation (+ sham tDCS), which slowed movement time specifically on right-cued targets (Hemifield x Time: p=0.007). Anodal tDCS further modulated this effect of prism adaptation (tDCS x Hemifield x Time: p=0.007), by speeding movement time overall (tDCS: p<0.001), with a stronger effect for right-cued targets (tDCS x Hemifield: p<0.001). Patient 3 only performed this task during the anodal intervention phase (i.e. no sham data), but with both arms. With the right arm (ipsilesional) overall movement time speed was increased after PA +M1 a-tDCS compared to before (Time: p=0.048), as in Patients 1 and 2. With the left (contralesional, recovered paretic) arm, prior to PA, Patient 3 could perform only 6/11 trials (left) and 4/11 trials (right) correctly. After PA +M1 a-tDCS, he could perform 10/11 (left) and 11/11 (right) trials successfully. Whether this improvement was an effect of the intervention, or simply of task repetition, is unclear. Thus, stimulation-augmented PA therapy did not impair motor performance in the left hemifield. Rather, the combination of PA +M1 a-tDCS (compared to sham) improved patients' overall reaching speed. Hence, the positive effects of stimulation-augmented prism therapy for visual neglect were accompanied by a motor facilitation effect on a task probe of normal everyday visually-guided reaching behaviour.

## Discussion

Our findings demonstrate that tonic disinhibition of left motor cortex during prism adaptation enhances consolidation of both sensorimotor and cognitive prism after-effects, stabilizing clinical gains in chronic, treatment-resistant visual neglect. This is a conceptually novel approach. Theoretical models of normal attention and neglect pathophysiology all emphasize that right brain injury and attentional dysfunction lead to pathological hyper-excitation of the left hemisphere, causing neglect (*Corbetta and Shulman, 2011*; *Kinsbourne, 1977*). When left hemisphere activity is suppressed, neglect improves, as seen during natural recovery (*Corbetta et al., 2005*), secondary lesions (*Vuilleumier et al., 1996*) and inhibitory brain stimulation (*Koch et al., 2008*). Our results in no way challenge the validity of these findings. However, here, by contrast, we targeted the left hemisphere with excitatory stimulation. The rationale to improve neglect was not based on conventional theoretical models, which prescribe 're-balancing' activity between the hemispheres, via excitatory stimulation of under-active (right) or inhibition of over-active (left) attention circuits (*Làdavas et al., 2015*; *Oliveri et al., 2001*; *Sparing et al., 2009*). A recent study aimed at such 're-balancing', by combining multi-session PA and parietal tDCS, did not obtain neglect improvements of the magnitude and longevity observed here with single-session PA + M1 a-tDCS (*Làdavas et al., 2015*). Instead, we aimed to induce adaptive functional plasticity, by tonically disinhibiting left M1 during adaptation, and thus potentiate the memory traces formed during PA. Critically, although located between hyper-excited attention structures in left parietal and frontal cortex, left M1 is an intact brain region in neglect, anatomically and functionally distinct from the direct, transcallosal and diaschitic effects

of the lesion (*Koch et al., 2008*). Thus we aimed not to directly regulate dysfunctional attention networks perturbed by the lesion, but rather to potentiate recruitment of functionally distinct, intact left sensorimotor circuits, whose engagement compensates for the lesion.

Initial scientific proof-of-concept evidence for efficacy of this experimental intervention was demonstrated in sham-controlled longitudinal case series in three patients with chronic treatment-resistant neglect (*Figure 7*, *Figure 7—figure supplement 1*). Patient 1 was tested with a phase reversal (A-B-B-A) design, while Patients 2 and 3 were tested with A-B/B-A designs. In all three patients, M1 a-tDCS during PA induced significant clinical gains in neglect that persisted throughout follow-up. Prior experimental work with single-session rehabilitation interventions for neglect has typically reported effects lasting ≤24 hr (*Barrett et al., 2012*), contrasting markedly with the endurance of benefit observed in all three patients here (18 to 46 days). Moreover, Patient 1 responded to both M1 a-tDCS interventions (Phase A and B; *Figure 7A,B*), and the effects accumulated, indicating that repeated interventions might further augment efficacy (for a related argument see *Panico et al., 2017*). The sustained leftward bias caused by PA +M1 a-tDCS that benefited visual neglect was not accompanied by (unintended) negative side effects for leftward motor behavior. A spatially cued reaching task (Appendix 2: *Supplementary file 2*) revealed that these patients exhibited 'directional hypokinesia'-like deficits, which are common in neglect (i.e. slower reach movement times when pre-cued to the left compared to the right). This left-right asymmetry was modulated by PA +M1 a-tDCS, which speeded reach movement times overall. Hence, the benefits for visual neglect were accompanied by facilitation of motor performance.

Importantly, these lasting gains for visual neglect occurred in chronic patients who were unresponsive to PA therapy (+ sham). This raises the question of whether PA or stimulation itself induced the clinical response. Our parsimonious interpretation is that PA is the therapeutic vehicle and M1 a-tDCS synergized this effect. There is a wealth of evidence that PA generalizes abnormally in neglect, transferring the leftward pointing bias into gains for visual cognitive deficits (*Jacquin-Courtois et al., 2013*). By contrast, we know of no evidence, cognitive mechanism, or anatomical pathway that could support the alternative hypothesis - that left M1 stimulation per se improved leftward visual attention. The control experiment of M1 a-tDCS during sham adaptation in healthy volunteers showed that stimulation alone induced no leftward shift in sensorimotor behavior (*Figure 2—figure supplement 1*). Importantly, all 3 patients reported here had shown improvements in neglect after PA therapy at the sub-acute stage. However, by the chronic phase tested here (1–1.5 years post-stroke) they were no longer responsive to prism therapy. In our clinical experience the emergence of treatment resistance between the sub-acute and chronic stage is not uncommon. How did M1 stimulation during PA reverse this? Our speculative interpretation is that M1 stimulation strengthened the developing leftward bias within adapting circuits during PA. By potentiating and perpetuating this bias over time, this somehow reactivated the cognitive transfer mechanism, unmasking latent responsivity in these formerly responsive patients, thus inducing a positive therapeutic effect. A non-specific arousal increase during stimulation cannot explain the gains in neglect, because stimulation was applied just once, 24 hours before the first post-intervention assessment.

tDCS can facilitate sensorimotor function in healthy volunteers and in stroke (*Hummel et al., 2005*; *Zimerman et al., 2013*). However, only when stimulation has been coupled with repetitive training or skill learning have effects previously been shown to out-last a single test session (*Reis et al., 2009*). Here, by contrast, we transformed the normally transient memory trace that follows a single session of PA into a long-lasting effect. How did this occur? From a computational perspective, we speculate that M1 stimulation changed how the sensorimotor system solves the temporal credit assignment problem imposed by errors during PA, biasing inference towards a long-timescale perturbation (*Kording et al., 2007*), and thus causing participants to update their internal models in a manner that gives rise to a long-lasting memory trace. This resulted in a persistent leftward shift in pointing accuracy, evident only in the context when visual feedback was deprived and participants had to rely on proprioception to guide reach accuracy (AE blocks).

The mechanism by which the leftward pointing bias induced by PA transfers to cognitive gains in neglect patients is not yet understood (*Jacquin-Courtois et al., 2013*). Presumably, the lesion disrupts the modularity of circuits that are normally functionally distinct (adaptation versus spatial cognition), causing abnormal transfer of the leftward bias across domains. The lack of a mechanistic understanding of the cognitive transfer process in neglect has made it difficult to devise a

principled means of maximizing the therapeutic efficacy of PA. Here we circumvented this problem, by targeting adaptation circuits, whose functioning is better understood. By leveraging knowledge of the role of M1 in motor memory formation, we intervened in a principled manner, aiming to enhance PA consolidation, and thus induced lasting pointing and cognitive prism after-effects. To achieve this, a mechanistic understanding of the cognitive transfer process in neglect was not required. Whatever that mechanism, here we demonstrate a principled method by which its effect can be dramatically enhanced. The present work describes a hypothesis-driven test of the prediction that M1-dependent PA consolidation mechanisms would enhance prism after-effects in the pointing and cognitive domains. The question of how other brain regions and mechanisms contribute to PA, or to cognitive transfer in neglect, was not addressed, and is beyond the scope of the present work.

The tDCS protocol used here is routinely applied in motor cortex studies (1mA, 20 min, 7 × 5 cm electrodes, anode over left M1, cathode over right eyebrow). However, the size of the anodal electrode means it extends to cover scalp locations overlying not only M1, but also dorsal premotor cortex and somatosensory cortex. In addition, the dipole montage distributes electric current throughout the intervening brain tissue between left M1 (anode) and the cathode. Given that the electric current distribution is therefore spatially diffuse, it is reasonable to question whether behavioural effects can be functionally localized to the motor cortex. While acknowledging the coarse spatial resolution of the induced electrical field, we maintain that such inference is reasonable. First, note that the same stimulation protocol (M1 a-tDCS) that augmented AE retention when applied during PA caused no such effect when applied at rest prior to the task (*Figure 2B, C*). This indicates that it is not motor cortex stimulation per se that is critical, but the concurrent interaction between a-tDCS and the functional (task) state of the brain that gives rise to the behaviour change. Second, although it is known that adaptation behaviour is implemented by distributed neural circuitry, with key causal nodes in parietal cortex, cerebellum and M1, only when a-tDCS was applied to M1 was retention enhanced. Given the likely connectional spread of tDCS between these three structurally interconnected regions (*Rivera-Urbina et al., 2015*), which are likely to functionally interact recurrently as adaptation progresses, implementing iterative processes such as error correction (*Hanajima et al., 2015*; *Inoue et al., 2016*), state estimation (*Miall et al., 2007*), sensory weighting and realignment (*Block et al., 2013*), etc. this finding of anatomical specificity supports the inference that M1 causally mediates early motor memory formation, consistent with previous work. Third, task performance imposes recurrent functional state transitions as adaptation progresses, and although M1 a-tDCS was applied throughout the Adaptation phase, the stimulation effect was functionally specific (*Figure 2A*, *Figure 3*). There was no effect on any trial in which visual feedback was present. There was no effect on error-dependent learning during Adaptation, nor on un-learning during Washout. Rather, there was a selective effect on after-effect persistence during Washout and Retention. After-effect trials require participants to rely on proprioceptive information to guide reach accuracy. This suggests that stimulation selectively enhanced the leftward proprioceptive-motor memory that developed during Adaptation. This is consistent with the anatomical location of the anodal electrode modulating recurrent functional interactions between the directly underlying motor-somatosensory cortices during adaptation. Fourth, our MRS experiment (*Figure 4*) provides an independent physiological measure that further grounds functional localization to M1. That experiment revealed covariation across individuals between the behavioural effect of M1 a-tDCS applied during prism adaptation and induced changes in M1 GABA concentration after M1 a-tDCS had been applied at rest. Thus, despite the fact that these two interference effects of M1 a-tDCS (behavioural, neurochemical) were measured weeks apart, in two different brain states (task versus rest), they nevertheless correlated in sign and magnitude across individuals. This suggests that intrinsic features of M1 physiology (responsiveness of the GABA system to modification) can account quantitatively for inter-individual variation in functional motor plasticity (see also *Stagg et al., 2011*). More specifically, the present data suggest that this physiology may causally constrain the impact of motor memory plasticity induction via M1 a-tDCS.

This study provides initial scientific proof-of-concept sham- and single case-controlled evidence that M1 a-tDCS can enhance the efficacy of prism therapy for neglect. Further extension in clinical testing should consider the following factors. The present findings were observed in chronic, severe, treatment-resistant patients who had been responsive to PA therapy at the sub-acute

stage. The effect of this intervention in patients with a different clinical profile remains to be investigated. Although it is well-replicated that, on average, M1 a-tDCS increases motor cortex excitability in healthy volunteers, stimulation effects vary across individuals. This has raised concerns about inter-individual variability and reliability of tDCS effects (*Buch et al., 2017*; *Jamil et al., 2017*). Our MRS experiment in healthy volunteers (*Figure 4*) demonstrates one way to address this question experimentally. Those results demonstrate that inter-individual variation in the behavioural effect of stimulation (enhanced AE retention) can be explained by inter-individual variation in the neurochemical response to tDCS. Individuals who showed the expected physiological response to excitatory stimulation (GABA decrease) showed a behavioural enhancement (retention increase). When GABA was unchanged by M1 a-tDCS, retention was also unchanged. Individuals in whom stimulation increased GABA showed a behavioural interference effect (AE retention decrease). Experiments of this nature inform about the underlying physiological sources of variability that can help explain behavioural variability in response to stimulation. Our MRS results suggest that physiological disinhibition of M1 is necessary (if not sufficient) for stimulation-enhanced PA consolidation to occur. Hence, we recommend incorporating a physiological marker of M1 response to stimulation in future work to help interrogate expected inter-individual variation in behavioural response. That stimulation increased GABA and disrupted AE consolidation in some individuals raises the possibility that stimulation could antagonize prism therapy in some patients. We therefore recommend cautious replication and extension of the present findings, via further case-controlled testing, in clinical application.

The present findings challenge consensus that because left fronto-parietal attention circuits are pathologically over-excited in neglect, the left hemisphere ought to be suppressed. Our data reveal latent plastic potential within intact left hemisphere sensorimotor circuits that can drive recovery if manipulated appropriately. Rather than suppress the left hemisphere, we aimed to maximize its adaptive capacity. We observed striking clinical gains by exploiting the paradox that rightward PA improves left attention, but only in the presence of a right hemisphere lesion. Such 'paradoxical functional facilitation effects' can restore or enhance function after brain injury (*Kapur, 1996*), offering an under-exploited opportunity to intervene experimentally to magnify neurologic gain. By exploiting this conceptual niche similar gains might be observed in other neurological domains (*Kapur et al., 2013*).

## Materials and methods

### Study design

The overall objective was a sham-controlled scientific initial proof-of-concept test of the *a priori* defined 1-tail hypothesis that M1 a-tDCS during prism adaptation (PA) would enhance consolidation of: (1) the leftward pointing bias (prism after-effect, AE); (2) the cognitive after-effect (neglect improvement). In addition, a magnetic resonance spectroscopy experiment was conducted as a physiological test of the hypothesis that the effect predicted in part 1 would depend causally on disinhibition (GABA decrease) in M1. Part 1 of the hypothesis was assessed, by measuring persistent leftward pointing errors after PA, in healthy volunteers under different brain stimulation conditions. The predicted effect of greater AE retention with anodal versus sham tDCS was confirmed (*Figure 2A,B*) and replicated in two additional experiments (*Figure 3*, *Figure 4—figure supplement 1*). Part 1 was also assessed in a single case patient study (Patient 1) comparing sham versus M1 a-tDCS using a phase reversal design (A-A-B-A) (*Figure 5*). Part 2 was assessed in three chronic stroke patients with severe treatment-resistant neglect by measuring longitudinal changes in behavioural performance across a battery of neglect tests (*Figure 7*). Patient 1 was assessed using a phase reversal design (A-B-B-A)(*Figure 7A,B*). Patients 2 and 3 were assessed using A-B and B-A designs (*Figure 7—figure supplement 1*). Part 3 was assessed in healthy volunteers by testing for predicted covariation between the physiological effect of stimulation (reduction in M1 GABA) and the behavioral effect (enhanced AE retention) (*Figure 4*). These three primary endpoints and the associated 1-tailed directional hypotheses were all specified *a priori*. All experiments used repeated measures designs with order counterbalancing and participants randomized to intervention order. The sample size of the healthy control experiments (n = 9/10) was determined based on expected adequacy to detect a significant anodal >sham effect. Since a previous adaptation/M1 tDCS study with n = 10

and a between-subjects design reported significant effects, the present repeated measures design was expected to have sufficient power (*Galea et al., 2011*). An *a priori* sample size calculation, based on the effect size reported in *Galea et al. (2011)*, confirmed this (for 1-tail t-test with effect size dz = 0.92, β = 80%, α = 0.5, required n = 9). *A posteriori* sensitivity analysis based on our observed results (sensitivity of 1-tail t-test with n = 9, β = 80%, α = 0.5, effect size dz = 0.9) further confirmed that our design and analysis strategy controlled Type 1 and 2 error rates appropriately. These statistical analyses were performed using G* power software (*Faul et al., 2007*). The patient sample size was chosen pragmatically within the constraint of limited access to chronic patients with severe treatment-resistant neglect and no contraindications to brain stimulation. Placebo control was achieved by using an established stimulation procedure with which neither patients nor healthy volunteers can reliably distinguish real from sham tDCS (*Gandiga et al., 2006*). Blinding efficacy was confirmed by post-test forced guessing (sham/real = 50:50) in healthy volunteers (experiment shown in *Figure 4*) and in patients (only 1 of 4 anodal/sham pairings was guessed correctly). In Part 1, outliers were defined as healthy volunteers whose mean baseline open-loop pointing performance (*Figure 1D*, step (2)) was shifted to the right (of zero). These were excluded prospectively because piloting showed that such individuals adapted but did not exhibit retention. One healthy volunteer was excluded retrospectively and replaced (*Figure 2D*) as he failed to adapt in the sham condition. In Part 2, Patient 2 forgot to attend the second baseline Sham session (*Figure 7—figure supplement 1*). Patient 2's score was calculated based on 8 out of 10 completed neglect tests. One test was omitted mistakenly from the baseline sham session, so was removed. A second test was omitted because the patient was highly distractible so it took too long to complete. Group analysis was therefore conducted across the eight subtests common to all patients. To enable repeated measures analyses, across the entire patient dataset reported here seven missing data-points (for Patient 1) were replaced by the average of 2 adjacent data-points on the same subtest in the same testing phase (*Figure 7—figure supplement 1*). In Part 3, 16/16 usable behavioural and 10/16 usable MRS datasets were acquired (1 participant did not attend the scan; 3 MRS acquisition errors owing to voxel misplacement; two unusable spectra owing to gradient coil overheating). All usable data were analysed (*Figure 4*, *Figure 4—source data 1*).

## Participants

Sixty-six right-handed healthy volunteers with normal or corrected-to-normal vision participated in this study (39 females; mean age = 26.6, SD = 7.5; 9 each in Experiments 1–4, 10 each in Experiments 5–7), in addition to three male chronic stroke patients (mean age = 58) with visual neglect. For detailed lesion and clinical information see SI. Lesion data are shown in *Figure 6*. All were screened for contraindications to tDCS. Informed consent was obtained after the nature and possible consequences of the studies were explained. Experiments in healthy volunteers were conducted in accordance with ethics approval at the University of Oxford (Oxfordshire REC A Ref: 06/Q1604, 13/SC/0163). The clinical phase was performed in Hôpital Henry Gabrielle, Lyon, in accordance with French laws governing clinical research (last version n°2004–806, 9th august 2004) and data protection (last version n°2004–801, 6th august 2004) abiding by the Declaration of Helsinki.

## Prism Adaptation

Participants sat at a table in a chinrest and pointed with the right hand in a centre-out reaching movement, upon instruction, to each of three circular targets, one located centrally (57 cm from eyes-to-target), one to the right (+10 cm) and one to the left (−10 cm). Pointing accuracy was measured on every trial as the lateral deviation of the reach endpoint from the target location (in cm).

There were two types of blocks: (A) *Closed-loop pointing (with visual feedback):* Participants made rapid pointing movements to the left and right targets in pseudo-random order. Continuous vision of the hand was available, thus providing dynamic error feedback to drive adaptation, except for the start position and the first ~30% of the reach trajectory, which were occluded by the chinrest, in order to minimize strategic (rather than adaptive) corrections of hand position. Both accuracy and speed were emphasized. See *Figure 2A* (Blocks E1-6, W1-6) for how closed-loop pointing accuracy changed across time. Pointing was accurate at baseline (*Figure 1D*: 1), and became right-shifted during the initial prism exposure period(*Figure 1D*: 3), but participants used visual error feedback to gradually correct their errors across trials until they re-gained baseline accuracy (*Figure 1D*: 4). In

the experiments shown in *Figure 2*, Adaptation was followed by an active washout phase, in which the prisms were removed, and participants made leftward pointing errors (a consequence of adaptation) (*Figure 1D*: 6). They again used visual error feedback to regain baseline accuracy (*Figure 2A*, Blocks W1-6). (B) *Open-loop pointing (no visual feedback):* Participants made naturally paced pointing movements at the central target, which was visible at reach onset, but was then occluded. Hence participants had to rely on proprioception to guide reach accuracy in the absence of visual feedback. The rationale for the occlusion of feedback is to enable adaptation after-effects to be assessed without the participant receiving that feedback and subsequently de-adapting. Accuracy was emphasized over speed. See *Figure 2A* (Blocks AE1-15) for how open-loop pointing accuracy changed across time. At baseline, open-loop pointing was less accurate than closed-loop pointing (*Figure 1D*: 1 versus 2). For each participant, the baseline mean pointing error was subtracted from all subsequent open-loop datapoints, ensuring that all measures of prism after-effect (AE) data represent a change (leftward error) from baseline. As participants adapted to the prismatic shift (*Figure 1D*: 3–4), open-loop pointing tended to shift leftward, ie: a prism after-effect emerged (*Figure 1D*: 5). AE magnitude was measured throughout Adaptation, Washout and Retention phases.

At baseline participants performed 20 trials of closed-loop pointing and 2 blocks of 15 trials of open-loop pointing. The Adaptation phase of each experiment consisted of 6 blocks of prism exposure (closed-loop: *Figure 2A*: E1-6; 100 trials: 10 in blocks 1–2, 20 in all subsequent blocks, equal number of trials to each of two target locations) alternating with 6 blocks of after-effect measurement (open-loop: *Figure 2A*: AE1-6; 90 trials: 15 per block). The Washout phase consisted of 6 blocks of closed-loop pointing with prisms removed (*Figure 2A*: W1-6; 60 trials: 10 per block) alternating with 6 blocks of AE measurement (90 trials: 15 per block). Retention consisted of 3 blocks of AE measurement (15 trials per block).

In experiments 1–4 (*Figure 2A–E*), participants underwent 3 test sessions of PA (Adapt, Washout, Retain protocol) combined with tDCS (sham, anodal, cathodal). Experiment 5 (*Figure 2—figure supplement 1*) had 2 sessions (anodal, sham) and used identical procedures except that neutral lenses were used. In experiments 6–7 (*Figures 3* and *4*), participants underwent 2 test sessions of PA (Adapt only protocol) combined with tDCS (sham, anodal), each followed by a retention test 24 hr later. In experiment 7 AE was sampled and averaged across the three target locations. All PA + tDCS sessions were separated by a minimum interval of 1 week. Stimulation order was counterbalanced across sessions.

## Transcranial Direct Current Stimulation (tDCS)

A DC-stimulator (Magstim, United Kingdom) delivered current via rubber electrodes (5 × 7 cm for M1 and PPC stimulation) (Easycap) fitted inside saline-soaked sponges and fixed to the head using rubber bands. The stimulation protocol (electrode montage and current intensity) for each of the three anatomical sites was derived from prior work (cited below) that had shown physiological or functional efficacy of that stimulation protocol over that brain region. For M1 stimulation, the active (anodal/cathodal) electrode was centred over the hand area of left primary motor cortex, 5 cm lateral to the vertex, with the reference electrode on the contralateral supraorbital ridge (*O'Shea et al., 2014a*). This location was confirmed by evoked hand muscle response to transcranial magnetic stimulation (for data in *Figure 2A–C*, *Figures 4,5,7A,B*). For left posterior parietal cortex, the active (anodal/cathodal) electrode was placed over electrode position P3 (of the international 10–20 system of EEG electrode placement), and the reference electrode was placed over Cz (*Sparing et al., 2009*). For right cerebellar stimulation, the active (anodal/cathodal) electrode was placed 3 cm lateral to the inion, with the reference electrode over the right buccinator muscle. In all conditions stimulation was applied at 1mA for 20 min, except for the cerebellum, as it has been shown that 2mA is required to induce sustained physiological effects (2mA, 20 min, 5 × 5 cm electrodes) (*Galea et al., 2009*). For real stimulation (anodal, cathodal), current was ramped up over 10 s, held constant for 20 min, then ramped down over 10 s. Sham stimulation used the same procedure except current was held constant for 30 s. With this procedure and 1mA M1 current neither healthy volunteers nor patients can reliably distinguish real from sham stimulation (*Gandiga et al., 2006*). This was confirmed by post-test forced guessing. For cerebellar stimulation (2mA), participants could distinguish real from sham, but not anodal from cathodal stimulation. Stimulation was well-tolerated by all participants. Reported side effects were restricted to a transient itch or tingling

sensation under one or both electrodes during current ramp-up, which dissipated over time. For stimulation in the scanner (*Figure 4*), custom electrodes (Easycap) were fitted with 5 kΩ resistors sited next to the electrode pad, to minimize eddy current induction in the leads causing heating under the electrodes. High chloride EEG electrode paste was used as a conducting medium between the scalp and the electrodes. MR-compatible extension leads connected the stimulator, located outside the magnetic field, to the participants. During the MRS acquisition the electrodes were unplugged from the stimulator.

## Neglect test batteries

The clinical phase employed two different batteries of standard paper-and-pen tests of neglect. The first battery was used only for Patient 1, phase 1 (*Figure 7A*) and comprised 6 tests: line bisection (mean of 10 samples), copy drawing, the Ota test (*Ota et al., 2001*), star cancellation (from the Behavioural Inattention Test, BIT [*Wilson et al., 1987*]), object cancellation (A4, dense search array) and object cancellation (A4, sparse search array). The second battery was used for all other test sessions (*Figure 7B–D*) and comprised 10 tests: letter cancellation (A3 paper), object cancellation (A3 sparse array), object cancellation (A3 dense array; *Figure 1C*), the Balloons tests (serial and popout visual search, A4)(*Edgeworth et al., 1998*), the Bells test (A3) (*Gauthier et al., 1989*), copy drawing, colouring, reading and filling in a mock administrative form. To minimize general learning effects, parallel versions of tests were used for copy drawing, object cancellation, colouring and reading tests in both batteries.

Performance on each test was scored out of 100%, starting from the right of the page (e.g. a patient who correctly completed only the right half of a test array would score 50%). Cancellation and search tests were scored as the percentage of targets correctly cancelled. Copy drawing and colouring tests consisted of 5 symmetrical items and were scored 10% per object drawn, with an additional 10% per symmetrical object drawn. Line bisection was scored as 100 minus the percentage rightward deviation from the centre of a 20 cm line. Reading was scored as the percentage of words correctly spoken aloud without errors or omissions. Filling in a mock administrative form was scored as the percentage of questions answered in writing, starting from the right. Percent correct scores across the subtests were then combined across all tests of the battery to compute the overall neglect test score for each patient. Scores were then combined across individuals for group mean figures and statistical testing.

## Magnetic Resonance Spectroscopy (MRS)

Healthy volunteers underwent two 3T MRI scan sessions in which MRS data (*Figure 4—source data 1*) were acquired before and after 20 min of 1mA a-tDCS was applied to left M1. In one session MRS data were acquired from a region centred on the hand knob of left M1 (*Figure 4B*); in the other (control) session data were acquired from occipital cortex, on an axial slice drawn through the anterior-posterior commissure line (*Figure 4—figure supplement 2*). Scan session order was counterbalanced and sessions were separated by at least one week. Participants lay awake and at rest in the scanner throughout 2 × 10 min MRS scans, acquired once each before and after tDCS. High-resolution T1-weighted anatomical images were acquired before and after the MRS acquisition and used to guide placement of the 2 $cm^3$ isotropic regions of interest for each individual. MRS data were acquired using the SPECIAL sequence (*Mekle et al., 2009*), which provides good reproducibility for measurements of GABA levels in the human brain (*Near et al., 2013*) Spectra were quantified using LCModel analysis software (*Provencher, 2001*). Estimates of neurotransmitter concentrations (GABA, Glutamate) were expressed relative to total Creatine (creatine +phosphocreatine), a commonly used reference that is easily detectable, and whose concentration remains relatively stable across individuals over time (*Turner and Gant, 2014*). To control for partial volume effects, neurotransmitter concentration estimates were corrected for grey and white matter tissue fractions in the measurement voxel (eg: [GABA:Cr * ((grey matter fraction +white matter fraction)/(grey matter fraction))], where total tissue = grey matter+white matter+cerebrospinal fluid), as reported previously (*Stagg et al., 2009*).

## Statistical analysis

In the experimental phase, the goal was to test the hypothesis that M1 a-tDCS during adaptation would enhance consolidation of the prism after-effect, measured as enhanced AE retention relative to sham. Hence, analyses contrasted relative AE magnitude in different stimulation conditions. To quantify AE for each individual, the endpoint error on each trial in each open-loop block (AE1-15) in each experimental phase (Adaptation, Washout, Retention) was normalized prior to analysis, by conversion to a change from the mean baseline open-loop score per individual. These delta scores were then averaged across trials within each block for each participant, and block means were analysed at the group level to test for stimulation effects. For the single case neglect patient study, AE magnitudes in each of the 15 trials in each of the 3 blocks of Retention were contrasted for each block across the four conditions (three sham, one anodal). In the clinical phase, patients' scores on each subtest of the neglect test battery were converted to percentages prior to analysis. For both the experimental and clinical phase, data were analysed using repeated measures (RM) ANOVAs and 1-tailed planned contrasts, to assess the *a priori* defined 1-tailed directional hypothesis that anodal tDCS (relative to sham) would cause: (1) larger AEs, specifically during Washout and Retention; (2) greater neglect improvement. *Post hoc* analyses in healthy volunteers confirmed no interaction between stimulation order and these hypothesis-driven contrasts (all p>0.05), confirming the effectiveness of order counterbalancing. For all group analyses of patient data the factor 'Patient' was modeled as a covariate, to account for individual differences and to avoid confounding intra- and inter-individual variance. Data distribution assumptions for RM ANOVA and t-tests of normality, heteroscedasticity of variance, and sphericity, were verified using Shapiro-Wilk, Levene's and Mauchly's tests, respectively, and sphericity violations were Huyhn-Feldt corrected. In the MRS experiment (*Figure 4*), Spearman's rank correlation coefficient tested the *a priori* defined 1-tailed hypothesis of a significant negative correlation between the M1 a-tDCS-induced GABA change (expected decrease) and AE retention change (expected increase). To avoid multiple comparison problems, only three correlation tests were performed: one test of this *a priori* prediction, plus one neurochemical (glutamate) and one anatomical (occipital GABA) control analysis. To assess relative AE retention 1 day after PA (for anodal versus sham), mean AE magnitude was quantified as a proportion of the AE achieved by the end of Adaptation the previous day (ie: proportional AE retention relative to mean of block AE6). Statistical significance was set at p=0.05, 1-tail for planned contrasts of Anodal versus Sham in the *a priori* defined key condition (M1 a-tDCS during Adaptation), and 2-tail for all other effects and interactions. The 1-tailed planned contrast (Anodal > Sham) was specified in advance within the same RM ANOVA model in which main effects and interactions were tested, using the 'simple' planned contrast procedure in SPSS, which contrasts an experimental condition against a control. Effect sizes were calculated for all *a priori* predicted effects (ie: anodal > sham) and are summarized in Appendix 2: *Supplementary file 1*. Simple main effect sizes were quantified using partial eta squared ($\eta^2_p$) for ANOVA and Cohen's *d* for planned contrasts. Cohen's *d* calculation was adjusted for repeated measures design (*Lenhard and Lenhard, 2014*). Analyses were conducted using SPSS version 23. Figures were plotted using Matlab (The Mathworks, Natick, MA) and show means ±1 SEM bars, adjusted for repeated measures designs (*Morey, 2008*).

## Acknowledgements

We thank Masud Husain, Chris Kennard and Glyn Humphreys for helpful comments on the manuscript and Jill X O'Reilly for assistance with figures.

## Additional information

### Competing interests

Heidi Johansen-Berg: Reviewing editor, *eLife*. The other authors declare that no competing interests exist.

## Funding

| Funder | Grant reference number | Author |
|---|---|---|
| Royal Society | Dorothy Hodgkin Fellowship | Jacinta O'Shea |
| Fondation pour la Recherche Médicale | Program urgency de recherche | Jacinta O'Shea<br>Yves Rossetti |
| National Institute for Health Research | Oxford Biomedical Research Centre | Jacinta O'Shea<br>Heidi Johansen-Berg |
| Institut National de la Santé et de la Recherche Médicale | Institutional support | Patrice Revol<br>Yves Rossetti |
| Hospices Civils de Lyon | Institutional support | Patrice Revol<br>Sophie Jacquin-Courtois<br>Gilles Rode<br>Yves Rossetti |
| Centre National de la Recherche Scientifique | Institutional support | Patrice Revol<br>Yves Rossetti |
| Cortex Université de Lyon | Labex/IdexANR-11-LABX-0042 | Patrice Revol<br>Yves Rossetti |
| European Commission | Marie Curie Initial Training Network | Pierre Petitet |
| Wellcome | Senior Fellowship | Heidi Johansen-Berg |

The funders had no role in study design, data collection and interpretation, or the decision to submit the work for publication.

## Author contributions

Jacinta O'Shea, Conceptualization, Formal analysis, Supervision, Funding acquisition, Investigation, Methodology, Writing—original draft, Project administration, Writing—review and editing, Designed, conducted and analysed all studies, interpreted the results, and wrote the paper; Patrice Revol, Investigation, Writing—review and editing, Carried out the kinematic task, assisted with neglect testing and assisted with preparing the manuscript; Helena Cousijn, Formal analysis, Investigation, Writing—review and editing, Contributed to data acquisition and analysis of the magnetic resonance spectroscopy experiment and assisted with preparing the manuscript; Jamie Near, Software, Formal analysis, Methodology, Writing—review and editing, Contributed to analysis of the magnetic resonance spectroscopy experiment and assisted with preparing the manuscript; Pierre Petitet, Investigation, Writing—review and editing, Carried out Experiment 5, analysed the patient lesion data, and assisted with preparing the manuscript; Sophie Jacquin-Courtois, Gilles Rode, Writing—review and editing, Facilitated patient participation, assisted with patient behavioural testing, and assisted with preparing the manuscript; Heidi Johansen-Berg, Funding acquisition, Writing—review and editing, Contributed to analysis of the magnetic resonance spectroscopy experiment and assisted with preparing the manuscript; Yves Rossetti, Conceptualization, Formal analysis, Funding acquisition, Investigation, Methodology, Writing—review and editing, Co-designed (with JO'S) the prism adaptation protocol and the clinical phase, contributed to acquisition and analysis of the patient data, and assisted with preparing the manuscript

## Author ORCIDs

Jacinta O'Shea http://orcid.org/0000-0002-6007-0698
Patrice Revol http://orcid.org/0000-0001-6063-6551
Pierre Petitet http://orcid.org/0000-0003-1422-5326
Yves Rossetti http://orcid.org/0000-0001-8867-4496

## Ethics

Human subjects: Informed consent was obtained after the nature and possible consequences of the studies were explained. Experiments in healthy volunteers were conducted in accordance with ethics approval at the University of Oxford (Oxfordshire REC A Ref: 06/Q1604, 13/SC/0163). The clinical phase was performed in Hôpital Henry Gabrielle, Lyon, in accordance with French laws governing

clinical research (last version n°2004-806, 9th august 2004) and data protection (last version n°2004-801, 6th august 2004) abiding by the Declaration of Helsinki.

## Decision letter and Author response
Decision letter https://doi.org/10.7554/eLife.26602.021
Author response https://doi.org/10.7554/eLife.26602.022

## Additional files

### Supplementary files
• Supplementary file 1. Summary of effect sizes for all *a priori* predictions in the paper. A > S refers to the predicted greater effect of M1 anodal tDCS (A) compared to sham (S). Each row describes the relevant figure, sample, 1-tailed hypothesis, statistical significance level, result (F, t, p) and effect size in both raw measurement units (mean difference, standard error of the mean, SEM, Spearman's r) and standardized units (partial eta squared $\eta^2_p$, Cohen's d), alongside published benchmark interpretations of the magnitude (*Sawilowsky, 2009*) and practical significance (*Wolf, 1986*) of Cohen's d effect sizes.
DOI: https://doi.org/10.7554/eLife.26602.016

• Supplementary file 2. Individual patients' reach kinematics before and after PA + tDCS. Patients 1 and 2 performed the task with the right arm once each in the days before and after PA was combined with M1 anodal or sham tDCS. Patient 3 underwent the anodal condition only, but with both the non-paretic (right) arm and the recovered paretic (left) arm. Table lists individual trial data plus means and standard deviation for all three patients. Accuracy data show the number of trials of the total (11) that were correct, were missed (no response to the 'go' cue), were too slow (outliers = RT and/or MT ± 2 SD of the patient's mean for that condition), or in which the patient subsequently placed the ball in the wrong basket (i.e. the left following a right pre-cue, and vice versa). Only the centre-out phase of the reach was analysed. PA combined with anodal tDCS (compared to sham) speeded centre-out reach movement time, regardless of whether the central pre-cue instructed a subsequent leftward or rightward movement after grasping the ball. Kinematic parameters characterize both the transport component [Reaction time (RT), Movement Time (MT), Peak Velocity (PV), Time to Peak Velocity (TPV), Maximal Height of Wrist and Elbow (MHW, MHE)] and the grasp component [Maximum Grip Aperture (MGA), Time of Grip Aperture Onset (TGA), Time of Maximum Grip Aperture (TMGA)] of the centre-out reaching movement required to grasp the tennis ball.
DOI: https://doi.org/10.7554/eLife.26602.017

• Transparent reporting form
DOI: https://doi.org/10.7554/eLife.26602.018

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

## Appendix 1

DOI: https://doi.org/10.7554/eLife.26602.019

# Supplementary materials and methods

## Magnetic resonance spectroscopy

Scans were performed on a 3T Siemens Verio scanner with a 32-channel head receive array. Short-TE localised single-voxel MRS data were acquired using the following scan parameters: TR/TE = 4000/8.5 ms, 2048 spectral points, and spectral width = 2000 Hz. 128 averages were performed using VAPOR water suppression, and an additional eight averages were collected without water suppression for referencing purposes. For all participants, a weighted recombination of 32-channel data was performed, and the SPECIAL inversion-on and inversion-off scans were subtracted, yielding 64 fully localized scans. B0 frequency and phase drift correction was performed using the MATLAB (The Mathworks, Natick, MA) -based FID-A toolkit (github.com/CIC-methods/FID-A) (*Simpson et al., 2017*) to ensure exact alignment prior to signal averaging. Processed MRS data were analyzed in LCModel using a simulated basis set consisting of 20 individual metabolites. To avoid bias, no metabolite concentration estimates were rejected based on cramer-rao lower bound (CRLB) uncertainty estimates, since this has been shown to selectively remove measurements with lower concentrations (*Kreis, 2016*). For M1 (*Figure 4*), CRLB values for GABA were 15.5 ± 2.9 (mean ± SD) and 22 (maximum) and the linewidth was. 029 ± 0.005 (mean ± SD). Corresponding values for occipital cortex (*Figure 4—figure supplement 2*) were: 15.5 ± 1.8 (mean ± SD) and 19 (maximum) and linewidth. 035 ± 0.008 (mean ± SD). The maximum linewidth across all datasets was. 055 Hz. Full details in *Figure 4—source data 1*.

The T1-weighted anatomical scan parameters were: 192 slices (1 mm thick), distance factor = 50%, fov(read)=200 mm, fov(phase)=90.6 mm, matrix = 174 × 192 × 192, TR = 2040 ms, TE = 4.68 ms, TI = 900 ms, flip angle = 8°, one concatenation, bandwidth = 130 Hz, echo spacing = 11.2 ms. FMRIB's Automated Segmentation Tool (FAST), part of the FMRIB software library (www.fmrib.ox.ac.uk/fsl), was used on the T1-weighted structural scan to calculate and correct metabolite concentrations for the relative quantities of grey matter, white matter and cerebrospinal fluid within the regions of interest.

## Patient clinical and lesion characteristics

All three patients had suffered a first ever right hemisphere stroke (*Figure 6*). Patient 1 was a 54 year old man who had suffered a right middle cerebral artery territory haemorrhagic stroke with widespread damage to the right perisylvian network. At the time of testing he was 1 year post-stroke and a long-term inpatient in the Lyon rehabilitation centre, Hôpital Henry Gabrielle. He received three physical and three occupational therapy sessions each week during the 6 month inclusion period. Importantly, Patient 1 was not in receipt of any rehabilitation for neglect during the period of participation in the study, and no change was made to his motor rehabilitation programme (physiotherapy). He had a severe, stable, chronic left neglect, as well as a complete left hemiplegia and complete left hemianopia. The lesion involved the superior parietal lobule, the anterior part of the inferior parietal lobule, the temporo-parietal junction, the superior and middle temporal gyri, the anterior part of the inferior temporal gyrus, the whole insula, and the lateral part of the frontal lobe including the precentral gyrus and the superior, middle and inferior frontal gyri. Subcortical white matter was also affected, as well as the internal capsule, putamen, centrum ovale and the rostrum and genu of the corpus callosum.

Patient 2 was a 57 year old man who had suffered a right haemorrhagic stroke affecting parieto-occipital cortex. At the time of testing he was 1.5 years post-stroke and was an outpatient in the Lyon hospital and so was not receiving any neglect rehabilitation. He had a severe, stable, chronic left neglect and complete left hemianopia. Initially hemiplegic, this had

recovered by the time of testing (*Figure 1B*). Patient 2's lesion involved the superior and inferior parietal lobules, the precuneus, the cuneus and posterior parts of the temporal lobe including the middle and inferior temporal gyri. Sub cortical white matter was also affected.

Patient 3 was a 63 year old man who had suffered a right capsulo-thalamic haematoma. At the time of testing he was 1.5 years post-stroke and was an outpatient in the Lyon hospital and so was not receiving any neglect rehabilitation. He had a severe, stable, chronic left neglect, as well as a complete left hemiplegia and complete left hemianopia. Patient 3's lesion did not affect any cortical structure. However, it affected the internal capsule, putamen, thalamus, centrum ovale and white matter.

To measure lesion overlap between the three patients (*Figure 6*), lesion masks were manually drawn on each patient's T1-weighted anatomical image and then registered to the MNI152 standard space using the non-linear registration tool (FNIRT), part of the FMRIB Software Library (FSL, http://fsl.fmrib.ox.ac.uk/fsl/fslwiki/FSL). Despite partial overlap between patients 1–2 and 1–3, there was no cortical or sub-cortical grey matter overlap between all three patients.

To identify any regions of white matter damage overlap across patients, Disconnectome map software, part of the BCB toolkit (http://www.brainconnectivitybehaviour.eu), was used (*Thiebaut de Schotten et al., 2014*). The registered lesion masks for each patient were used as seedpoints to identify which tracks passed through the damaged regions, by comparison with a reference dataset of healthy controls. This technique allowed us to identify the white matter tracts that were likely to be interrupted by the lesion. Each patient's disconnectome map was thresholded at a value of $p > 0.9$ and then overlaid on each other in order to identify the tracts most likely to be disconnected in all three patients. This analysis revealed three white matter tracts: the right inferior fronto-occipital fasciculus (rIFOF), the right superior longitudinal fasciculus (rSLF) and the right anterior thalamic radiation (rATR) (*Thiebaut de Schotten et al., 2011*; *Thiebaut de Schotten et al., 2005*).

## Naturalistic spatial reaching task

Each patient was seated in a chair in front of the experimental table. The task was to reach out and grasp a tennis ball (diameter 6.5 cm) located 35 cm in front of the start point, aligned on the body midline, and to place the ball in either a left or a right basket (centred 20 cm lateral of the ball position), depending on a visual instruction cue. Each trial began with the right hand on the starting point, located 5 cm in front of the body midline, with the thumb and index finger opposed, and with the forearm resting on the table. A central 4 s alert signal (flashing red dot) triggered the start of the trial. This was followed immediately by a central 3 s 'go signal' (green arrow) that instructed the patient to place the ball subsequently in the left or right basket after it had been grasped. Data acquisition began at 'go signal' onset. Kinematic parameters were computed and analysed only for the first phase of the movement i.e. the centre-out reach-to-grasp the tennis ball. Note therefore that the movement was identical on each trial, so the distinction between 'left' and 'right' trials was only that patients were cued in advance as to which of a leftward or rightward movement they would have to perform after they had grasped the ball. Movements were self-paced (i.e. no experimenter-imposed speed instructions), and outlier trials were defined as those in which reaction time and/or movement time exceeded the patient's mean ±2 SD for that condition. These were removed prior to analysis (*Supplementary file 2*). We expected this task to be sensitive to neglect, since "directional hypokinesia" - impairment of contralesionally directed (i.e. leftward) movements - is a common symptom of neglect patients. Hence, at baseline, we expected patients to show a relative impairment (slowing) on left-cued trials compared to right. Statistical analyses asked whether PA and/or M1 a-tDCS would alter this expected pattern of behaviour. In each test session patients performed 22 trials (11 left, 11 right) in a pseudo-randomized order over ~15 min. All movements were self-paced and performed without prior training.

Movement of the right upper limb was recorded by an optoelectronic motion capture system (MX Giganet, Vicon, Oxford, UK), composed of six infrared stroboscopes and infrared cameras at 100 Hz sampling frequency. To track limb position over time, four passive infrared

reflecting markers were placed upon the nail of the thumb and index finger, the styloid process of the radius at the wrist, and the lateral epicondyle at the elbow. After recording and tri-dimensional reconstruction, the position data of each marker were filtered with a Butterworth filter, with a cut-off frequency of 6 Hz.

Patients 1 and 2 performed one session before and after each sham and anodal intervention. Patient 3 performed the task only before and after the anodal session, but with both arms, so his non-paretic (right) and recovered paretic (left) arm could both be assessed. *Supplementary file 2* lists the kinematic parameters calculated for each trial of each individual patient in each test session before and after PA + tDCS. Reaction Time (RT, in ms) was calculated as the delay between the go signal and the release of the finger and thumb from the start position. Movement Time (MT, in ms) was computed as the time between the onset and offset of the movement, i.e when a stable grip on the ball was achieved. Peak Velocity (mm/s) and the Time to Peak Velocity (ms) of the wrist were calculated with respect to movement onset. The maximal height of the wrist and elbow above the table (MHW and MHE, in mm) were also calculated. For the grasp component of the movement, the Maximum Grip Aperture (MGA, in mm) was calculated as the maximal distance between thumb and index finger during the reach. Finally, latencies of the grip aperture onset (TGA, in ms) and time to reach maximum grip aperture (TMGA in ms) were computed with respect to movement onset.

# Appendix 2

DOI: https://doi.org/10.7554/eLife.26602.020

## Supplementary Text

### Results of Statistical Analyses

For a summary of statistical results and effect sizes for all predicted effects see *Supplementary file 1*.

### M1 stimulation enhances or impairs prism after-effect depending on current polarity and cognitive state

Experiments 1 and 2 (*Figure 2A–C*) aimed to test the prediction that M1 anodal tDCS would: (i) enhance AE consolidation, in a selective manner that depended both on: (ii) stimulation polarity (anodal but not cathodal); and (iii) cognitive state (tDCS during PA but not during rest prior to PA). Two groups of participants (n = 9 per experiment) underwent PA and tDCS in a repeated measures design, in three separate sessions, at least 1 week apart, in which they received anodal, cathodal or sham tDCS to M1. One group received stimulation while at rest before PA, the other during PA. RM ANOVA contrasted the between-subjects factor of cognitive state (stimulation before versus during PA) with the within-subjects factors of tDCS (anodal, cathodal, sham) and Block (AE $_{1-6}$, AE $_{7-12}$, or AE $_{13-16}$), separately for each experimental phase (Adaptation, Washout, Retention). Results are plotted in *Figure 2B and C*.

### Adaptation Phase (Blocks AE$_{1-6}$)

AE magnitudes stabilized progressively over time (Block $_{AE1-6}$: $F(1.714, 27.43)=13.393$, $p<0.001$) as a function of interleaved blocks of prism exposure, an overall pattern that did not differ between the two groups (Block × State: $F(1.714, 27.43) = 1.148$, $p=0.342$; *Figure 2B* versus 2C, Block $_{AE1-6}$). Importantly, however, a significant State × tDCS interaction ($F(2,32) =5.591$, $p=0.008$; $\eta^2_p = 0.259$) indicated that the effect of stimulation varied with cognitive state. Planned contrasts across cognitive state confirmed that, relative to sham, the behavioural impact of both anodal ($F(1,16)=10.795$, $p=0.005$; $\eta^2_p = 0.403$) and cathodal ($F(1,16)=4.606$, $p=0.048$; $\eta^2_p = 0.224$) tDCS differed significantly depending on whether stimulation was applied before or during the Adaptation phase. No other effects were significant.

For the group that had stimulation *during* Adaptation (*Figure 2A,B*), the main effect of tDCS was not significant ($F(2,16)=1.84$, $p=0.191$), although planned contrasts against sham revealed a tendency for anodal stimulation to increase AE magnitudes during Adaptation $_{AE1-6}$ ($F(1,8)=3.862$, $p=0.042$, 1-tail; $\eta^2_p = 0.326$; mean difference: 0.697, SEM: 0.354; Cohen's d = 0.504), whereas cathodal stimulation had no effect ($p>0.6$).

By contrast, in the group that had tDCS *prior* to Adaptation (*Figure 2C*), AE magnitudes during Adaptation $_{AE1-6}$ were reduced (tDCS: $F(2,16)=5.143$, $p=0.019$; $\eta^2_p = 0.391$), both by anodal ($F(1,8)=7.135$, $p=0.028$; $\eta^2_p = 0.471$) and cathodal stimulation ($F(1,8)=9.570$, $p=0.015$; $\eta^2_p = 0.545$).

### Washout Phase (Blocks AE$_{7-12}$)

After Adaptation, the prisms were removed, and participants were required to point to the left and right target, thus exposing leftward errors acquired by Adaptation to active washout. Participants corrected these errors rightward to restore baseline accuracy across successive blocks spaced out over 20 min, with interleaved blocks assessing the consequent AE washout rate. The key experimental prediction was that M1 a-tDCS would cause the AE to resist active washout, specifically when stimulation had been applied *during* but not *before* the Adaptation phase.

AE magnitudes decayed gradually over time (Block$_{AE7-12}$: $F(2.967,47.467) = 9.642$, $p<0.001$) as a function of interleaved blocks of error correction, an overall pattern that did not differ

between the groups (Block × State: F(2.967,47.467) = 0.261, p<0.9; *Figure 2B* versus 2C, Block $_{AE7-12}$). The impact of tDCS on AE washout varied significantly with cognitive state (State ×tDCS: F(2,32)=6.759, p=0.004; $\eta^2_p$ = 0.297). Planned contrasts versus sham revealed that the cognitive state dependence of the stimulation effect was specific to the anodal (F(1,16)=12.083, p=0.003; $\eta^2_p$ = 0.430) but not cathodal condition (p>0.2). All else was n.s.

For the group that underwent stimulation *during* Adaptation (*Figure 2A,B*), there was a main effect of tDCS on AE washout (F(2,16)=4.555, p=0.027; $\eta^2_p$ = 0.363), with planned contrasts confirming the key prediction: less washout with M1 anodal tDCS relative to sham (F(1,8)=9.054, p=0.008, 1-tail; $\eta^2_p$ = 0.531; mean difference: 1.259, SEM: 0.418; Cohen's d = 0.944), a large effect that was polarity-specific (cathodal: p>0.4). To control for the larger AEs in the Adaptation phase caused by anodal tDCS, proportional washout was also assessed. RM ANOVAs on these ratio scores further verified reduced washout. Hence, M1 anodal tDCS during Adaptation increased the durability of the AE memory trace, indexed by slowed subsequent proportional forgetting.

By contrast, for the group that had stimulation *prior* to Adaptation (*Figure 2C*), there was no significant effect on AE washout (tDCS: F(2,16)=2.556, p=0.109). Although planned contrasts with sham suggested a tendency toward greater washout in the anodal condition (F(1,8)=4.284, p=0.036, 1-tailed; cathodal p>0.36), ratio analyses indicated there was no additional change in AE magnitude once the reduction already observed during the Adaptation phase was controlled for.

### Retention Phase (Blocks AE$_{13-15}$)

Following Washout, and a 10 min rest period, AE retention was assessed. The same pattern of cognitive state dependence and polarity specificity was observed (State × tDCS: F(2,26)=4.605, p=0.019, $\eta^2_p$ = 0.262; anodal tDCS versus sham: F(1,13)=6.258, p=0.027, $\eta^2_p$ = 0.325; cathodal p>0.6). All else was n.s.

For the group that had stimulation *during* Adaptation (*Figure 2A,B*), although the main effect of tDCS was not significant (F(2,12)=2.919, p=0.093, $\eta^2_p$ = 0.327), planned contrasts revealed significantly greater AE retention selectively with M1 anodal tDCS compared to sham (F(1,6)=4.941, p=0.034, 1-tail, $\eta^2_p$ = 0.452, mean difference: −0.969, SEM: 0.429, Cohen's d = 0.911; cathodal = p > 0.8). A one-sample t-test (versus 0) confirmed a significant leftward pointing bias persisted at retention, but only in the anodal tDCS condition (t(6)=2.996, p=0.024). By contrast, the AE had already washed out completely in both the sham (p>0.4) and cathodal (p>0.6) conditions. The same analyses with retention ratios verified this pattern.

By contrast, for the group that had stimulation *prior* to Adaptation (*Figure 2C*), there was no effect of tDCS on retention (F(2,14)=2.355, p=0.131, $\eta^2_p$ = 0.252; all planned contrasts: p>0.12), reflecting the fact that the AE had already washed out completely in all conditions (one-sample t-test: all p>0.25).

In summary, Experiments 1 and 2 confirmed the key prediction: relative to sham, M1 plasticity induction, via anodal tDCS applied *during* Adaptation (*Figure 2A,B*) potentiated memory, expressed as reduced washout and enhanced retention of the prism after-effect. By contrast, the same stimulation applied *prior* to adaptation (*Figure 2C*) tended to antagonize prism after-effects. Hence M1 plasticity induction potentiated AE consolidation in a manner that was polarity-specific and cognitive state dependent.

## No effect of tDCS on pointing accuracy when visual feedback is available

It is possible that the larger, persistent prism after-effects observed with M1 anodal tDCS during Adaptation could have arisen, not by tDCS-enhanced consolidation of AE, but as a consequence of tDCS changing the rate of error correction (eg: larger AEs from faster error correction during Adaptation, and/or slower error correction during Washout). To address this, linear slopes were fitted per participant to quantify the error correction rates in each block. Stimulation did not change the rate of leftward error correction during prism exposure (Block $_{E1-6}$, tDCS = p < 0.8, tDCS x Block = p < 0.9), nor did it alter the rate of subsequent

rightward error correction after prism removal (Block $_{W1-6}$: tDCS = p < 0.7, tDCS x Block = p < 0.9)(**Figure 2A**, blocks E1-6, W1-6).

Strikingly, during the Washout phase, during error correction blocks in which visual feedback was available (**Figure 2A** W1-6), M1 anodal tDCS had no effect on pointing accuracy. This contrasts with the interleaved blocks of AE measurement (**Figure 2A**, AE7-12), in which visual feedback was deprived, and participants exhibited a persistent leftward pointing error. Hence, the expected coupling between the magnitude of open-loop error at the end of a block (AE7-12) and the magnitude of closed-loop error at the start of the subsequent block (W1-6), while present in the sham condition, was uncoupled by anodal tDCS. The key difference between closed and open-loop conditions is the presence or absence of visual feedback. During closed-loop pointing, both visual and proprioceptive error feedback can be used to guide pointing accuracy. In the open-loop condition, only proprioception is available. The striking appearance and disappearance of a leftward shift in pointing across successive interleaved blocks of open- versus closed-loop pointing during Washout suggests that tDCS-enhanced consolidation is a proprioceptive effect. Consistent with the known dominance of vision over proprioception (**van Beers et al., 1999**) the data indicate that when visual feedback is available to guide pointing accuracy it is sufficient to compensate fully for the stimulation-induced persistent leftward proprioceptive bias.

Thus, the effect of M1 a-tDCS was functionally specific: it enhanced AE consolidation with no effect on error correction. M1 anodal tDCS during Adaptation had no effect on visually-guided error correction, neither during Adaptation nor Washout, but durability of the AE memory trace was selectively strengthened, an effect measurable only during epochs when participants had to rely on proprioceptive (but not visual) feedback to guide reach accuracy (ie: AE7-15).

## M1 a-tDCS–enhanced AE consolidation is anatomically specific

To determine the anatomical specificity of the M1 a-tDCS consolidation effect, the identical experiment was performed, except that stimulation was applied during the Adaptation phase to left posterior parietal cortex (PPC) (Experiment 3, n = 9) and right cerebellum (Experiment 4, n = 9). Statistical analyses focused on testing whether enhanced AE persistence was a specific consequence of stimulating left motor cortex, or whether the same effect could be induced by stimulation of left parietal cortex or right cerebellum.

Unlike for left M1, left PPC stimulation did not significantly change AEs in any experimental phase (tDCS: all p>0.1; **Figure 2D**). In directly contrasting the effect of M1 and PPC stimulation (**Figure 2B** versus 2D), while the overall contrast across brain regions was not significant (Region $\times$tDCS, all p>0.06), the more specific planned contrast of the anodal AE potentiation effect (anodal versus sham, 1-tail) across brain regions (PPC versus M1) was significant, during both the Washout $_{(AE7-12)}$ (F(1,16)=4.465, p=0.025, $\eta^2_p$ = 0.218) and Retention $_{(AE13-15)}$ phases (F(1,14)=3.342, p=0.044, $\eta^2_p$ = 0.193), although not during Adaptation $_{(AE1-6)}$ (F(1,16)=1.813 p=0.098, $\eta^2_p$ = 0.102). The same contrasts for cathodal tDCS were not significant (all p>0.2).

The same analysis approach compared the effect of M1 versus cerebellar tDCS. Unlike for left M1, but similar to left PPC, right cerebellar stimulation did not significantly change AEs in any experimental phase (tDCS: all p>0.2; **Figure 2E**). In directly contrasting the effect of M1 and cerebellar stimulation (**Figure 2B** versus 2E), the overall contrast across brain regions was significant during Washout (Region $\times$tDCS: F(2,32)=4.134, p=0.025, $\eta^2_p$ = 0.205), but not during Adaptation or Retention (all p>0.1). The more specific planned contrast of the anodal AE potentiation effect (anodal versus sham, 1-tail) across brain regions (cerebellum versus M1) was significant, during both the Washout $_{(AE7-12)}$ (F(1,16)=5.894, p=0.027, $\eta^2_p$ = 0.269) and Retention $_{(AE13-15)}$ phases (F(1,15)=4.404, p=0.026, 1-tail, $\eta^2_p$ = 0.227), although not during Adaptation $_{(AE1-6)}$ (F(1,16)=1.744 p=0.2). The same contrasts for cathodal tDCS were not significant (all p>0.1).

Hence, two control experiments confirmed that enhanced AE consolidation, as reflected in larger AEs during Washout, was an anatomically specific consequence of M1 a-tDCS. No such effect was observed with stimulation of left PPC or right cerebellum.

## No effect of M1 a-tDCS when applied during sham adaptation

To test whether anodal tDCS alone could induce the sustained leftward shift in pointing behavior that occurs when PA is combined with anodal (but not sham) tDCS during Adaptation, control Experiment five was performed. Identical procedures were followed as for all experiments in *Figure 2*, except that sham glasses (not prisms) were used. Data are shown in *Figure 2—figure supplement 1*. There was no leftward shift in pointing behavior when (sham) PA was performed, regardless of whether sham or real tDCS was applied during sham adaptation. Statistical analysis showed there was no significant difference between the two conditions (all p>0.3). Pointing accuracy in both conditions drifted rightward over time, likely reflecting boredom from repeated pointing over an hour at the same targets without any prismatic shift. A rightward drift in behavior with increasing sustained attention demand has been reported previously (*Manly et al., 2005*). If this factor also affected the experiments using real prism adaptation, this may have reduced the magnitude of the measured leftward shift. The results of this control experiment confirm that M1 anodal tDCS by itself does not induce a sustained leftward shift in sensorimotor behavior. Rather, this phenomenon arises instead specifically when M1 a-tDCS is applied concurrent with PA.

## M1 a-tDCS-enhanced AE consolidation persists across days

To determine the timescale of M1 a-tDCS-enhanced AE consolidation, Experiment 6 (n = 10) assessed relative retention of the prism after-effect over five days after sham or M1 anodal tDCS (*Figure 3*). RM ANOVA assessed the within-subjects factors of tDCS (anodal, sham) and Block and the between-subjects factor of Order (A-S versus S-A).

During Adaptation, a similar pattern was observed as in Experiment 1 (*Figure 2A*): compared to sham, there was no effect of M1 a-tDCS on error correction (Block $_{E1-6}$). AEs were larger in the anodal condition compared to sham (Block $_{AE1-6}$: tDCS: $F(1,8)=4.093$, $p=0.039$, 1-tail, $\eta^2_p = 0.338$, Cohen's d = 0.355), and this effect was larger if the anodal condition occurred first (tDCS x Order: $F(1,8) = 16.405$, $p=0.004$; $\eta^2_p = 0.672$).

Prism after-effect persistence was assessed across five retention intervals (10 min, 1, 2, 3, 4 days) using RM ANOVA with factors of tDCS, Time and Order. After prism removal, AE magnitudes decayed progressively (Time: $F(4,32)=10.739$, $p<0.001$), but this was significantly slowed by anodal stimulation (tDCS: $F(1,8)=15.857$, $p=0.002$, 1-tail, $\eta^2_p = 0.665$; mean difference: 1.03, SEM: 0.245, Cohen's d = 1.263; no interaction with Block or Order p<0.9), indicating a large increase in AE retention with M1 a-tDCS relative to sham. To control for the tDCS x Order interaction during Adaptation, the same analysis was conducted on AE data transformed into retention ratios (ie: ratio of the mean AE in the final block, $AE_6$). The same pattern was observed (Time: $F(4,32)=10.239$, $p<0.001$ and tDCS: $F(1,8)=3.292$, $p=0.053$, 1-tail, $\eta^2_p = 0.292$; mean difference: 17.532, SEM: 9.663, Cohen's d = 0.791; no interaction with Time or Order p<0.5).

To verify whether the sustained leftward pointing bias observed after PA +M1 a-tDCS was truly outside the normal range of participants' day-to-day pointing variability, we compared mean pointing accuracy (raw scores) over 4 days prior to PA (baseline) with the four days after each PA session (anodal and sham) (*Figure 3—figure supplement 1*). The effect of tDCS (baseline, post-anodal, post-sham) was significant ($F(2,16)=4.656$, $p=0.045$ $\eta^2_p = 0.350$; interaction of tDCS x Time p<0.8), and planned contrasts against baseline confirmed the presence of a significantly left-shifted error across days after PA +M1 a-tDCS ($F(1,9)=17.116$, $p<0.001$, 1-tail, $\eta^2_p = 0.655$, Cohen's d = 0.73), whereas accuracy was no different from baseline after PA +sham tDCS (p<0.5, $\eta^2_p = 0.042$).

In summary, M1 a-tDCS enhanced AE consolidation, resulting in a sustained prism after-effect that persisted across days.

## Covariation between the behavioural and neurochemical effects of M1 a-tDCS

To test the hypothesis that stimulation-enhanced long-term AE retention depends causally on M1, Experiment 7 tested for a correlation across individuals between M1 a-tDCS-induced changes in: a) M1 cortical inhibitory tone (GABA concentration) and b) AE retention.

16 participants underwent 2 PA sessions (separated by at least one week) in which tDCS (anodal/sham) was applied during Adaptation and retention was assessed after a short interval (10 min) and a long interval (24 hr). Participants underwent 2 MRS scans to quantify change in metabolite concentrations (GABA, Glutamate) after M1 a-tDCS (20 min, 1mA) - 1 session quantified metabolites in M1, and a control session quantified metabolites in occipital cortex. Analyses were conducted on the 16/16 usable PA/tDCS datasets and 10/16 usable MRS/tDCS datasets (see Materials and methods: Study Design).

There was no significant change in mean M1 GABA concentration after anodal tDCS (t(9) =.459, p=0.328, 1-tail, Cohen's d = 0.135). Behaviourally, during Adaptation a similar pattern was observed as in Experiment 5 (*Figure 3*): while the main effect of tDCS on AEs was not significant (p>0.5), there was a significant interaction of tDCS with Order, indicating larger AEs with anodal versus sham tDCS when the anodal condition occurred first (tDCS x Order: F(1,14) = 4.62, p=0.05; $\eta^2_p$ = 0.248). This interaction likely explains the lack of a main effect of tDCS on short or long-term retention (tDCS = 10 min interval: p<0.212, 1-tail; 24 hr interval: p=0.099, 1-tail). As in Experiment 5, to control for this tDCS x Order interaction during Adaptation, the retention analysis was run on AE data transformed into retention ratios. AE retention was quantified for each individual for each PA session, anodal and sham, as the proportional AE that persisted after PA, relative to the maximal AE achieved by the end of Adaptation (ie: ratio of the mean AE in the final block, $AE_6$). Logically, this quantity is bounded between 0-100%, reflecting minimal and maximal retention. Planned contrast of anodal-sham retention ratios confirmed M1 a-tDCS-enhanced AE short-term retention (10 min interval: t(15) =2.283, p=0.018, 1-tail, Cohen's d = 0.625) and long-term retention (24 hr interval: t(15) =2.068, p=0.028, 1-tail, Cohen's d = 0.658) (*Figure 4—figure supplement 1*).

To test the key hypothesis of co-variation between the neurochemical and behavioural effects of stimulation, we computed the correlation across individuals between the M1 a-tDCS-induced change in long-term AE retention (anodal-sham) and M1 GABA concentration (post-pre) (*Figure 4—source data 1*, *Figure 4—figure supplement 1*). Spearman's correlation coefficient confirmed a significant negative relationship ($r_s$ (10)=-.83, p=0.0015, 1-tail) (*Figure 4*). That is, the greater the GABA reduction with M1 a-tDCS, the greater the increase in relative AE retention. Conversely, those who did not show the GABA reduction did not show an increase in AE retention. Control analyses confirmed there was no correlation between the change in percent AE retention and glutamate change in M1 ($r_s$ (10)=0.297, p=0.405), and no correlation with GABA change in occipital cortex ($r_s$ (10)=-.382, p=0.276). The direct contrast of these correlation coefficients using Fisher's r to z transformation confirmed that the AE x M1 GABA change correlation differed significantly from the same correlation with M1 Glu change (z = 1.874 p=0.03) and occipital cortex GABA change (z = 1.642, p=0.05), demonstrating both neurochemical and regional specificity of this effect. AE retention ratios and changes in GABA concentration were converted to percent scores for plotting.

## Clinical phase

### Stimulation enhances AE retention in a single case neglect patient

To test the hypothesis that anodal tDCS applied during Adaptation would enhance AE retention also in neglect, Patient 1 underwent the behavioural protocol shown in *Figure 2* (Adaptation, Washout, Retention) on four occasions over 4 months. Only on the third occasion was true anodal stimulation applied, the other 3 sessions used sham (S-S-A-S). *Figure 5* shows

AE data for the anodal and sham sessions, the latter plotted with 95% confidence intervals on the patient's typical behaviour (computed across the mean of the 3 sham sessions). To first test whether a significant AE was still present at Retention (ie: sustained leftward shift in pointing relative to baseline), one-sample t-tests (against zero) compared AE magnitudes in each of the 3 retention blocks (AE13-15). This was performed on the 15 trials per block for anodal and on the mean across the 3 sessions for sham. This confirmed a significant AE was retained in both conditions (all $t_{(14)} > 3.472$, all $p < 0.004$).

To test the prediction that increased AE retention would occur with anodal versus sham tDCS, RM ANOVA and planned contrasts were computed across the 15 trials in each of the 4 test sessions (i.e. 3 sham, 1 anodal), separately for each of the 3 blocks of retention (AE13-15). There was a significant effect of Session (S-S-A-S)($F_{(3,42)} = 28.611$, $p < 0.001$; $\eta^2_p = 0.671$), Block ($F_{(2, 28)} = 16.835$, $p < 0.001$; $\eta^2_p = 0.546$) and their interaction ($F_{(6,84)} = 3.908$, $p < 0.013$). Follow-up ANOVAs compared the effect of anodal versus sham separately in each of the 3 blocks of retention. There was significantly enhanced AE retention with anodal tDCS versus sham in block 1 (AE13: $F_{(3, 42)} = 8.974$, $p < 0.001$; $\eta^2_p = 0.391$), block 2 (AE14: $F_{(3, 42)} = 19.115$, $p < 0.001$; $\eta^2_p = 0.577$) and block 3 (AE15: $F_{(3, 42)} = 50.097$, $p < 0.001$; $\eta^2_p = 0.782$). Planned contrasts within each block compared each sham condition against the anodal condition. All were significant (all $F_{(1, 14)} > 3.1$, all p range from $< 0.045$ to $< 0.001$, 1-tail; all $\eta^2_p$ range from 0.184 to 0.883), except for the contrast of the second sham session in block AE14 ($p > 0.5$). The contrast of mean anodal versus mean sham (sessions averaged within and then across blocks) yielded Cohen's d = 2.382. Hence, anodal tDCS applied during Adaptation increased AE retention relative to sham in this single case neglect patient.

## Stimulation causes durable gains from PA therapy in resistant neglect

To test whether M1 a-tDCS during PA would enhance the clinical efficacy of PA therapy, three patients with chronic, treatment-unresponsive neglect underwent serial longitudinal neuropsychological assessments before and after PA combined with sham versus M1 a-tDCS during adaptation.

For data see *Figure 7—source data 1*. Statistical analyses assessed the efficacy of PA + tDCS both at the individual patient level and the group level. RM ANOVAs and planned contrasts (1-tail) tested the *a priori*-defined 1-tail directional hypothesis that there would be a greater improvement in neglect score after versus before PA +M1 a-tDCS compared to sham.

Over a 9 month period, Patient 1 underwent a total of six sessions of PA +M1 a-tDCS. Sessions 1–4 consisted of Adaptation, Washout and Retention phases, 3 sham and 1 anodal (in order anodal/sham S-S-A-S), which were analysed to test for the stimulation-enhanced AE consolidation effect shown in *Figure 5*. For sessions 3–6, neglect was assessed over several weeks before and after each PA + tDCS intervention, in two phases (Phase A, A-S; phase B, S-A, across 211 days)(*Figure 7A,B*).

In Phase A (102 days, sessions 3, 4), neglect was assessed with a battery of six neuropsychological tests, four times in the two weeks preceding PA, and five times over the subsequent three weeks. During Phase A, Patient 1 underwent the Adaptation-Washout-Retention PA procedure, in which M1 a-tDCS had enhanced his retention of the prism after-effect (*Figure 5*). After PA combined with M1 a-tDCS, Patient 1's neglect improved by ~20% (~50% relative gain)(*Figure 7A*), a gain that had not decayed by the latest post-test, 21 days later. Statistics confirmed a significant effect. RM ANOVA on percent neglect scores across each of the six subtests revealed a significant change in neglect score over Time (6 levels: mean pre-PA versus 5 post-tests; $F_{(5,25)} = 5.632$, $p = 0.001$; $\eta^2_p = 0.53$), and the planned contrast (mean post-pre) confirmed there was a significant improvement in neglect after PA + anodal tDCS ($t_{(5)} = -4.054$, $p = 0.005$, 1-tail; mean difference: $-14.699$, SEM: 3.62; Cohen's d: 0.578). By contrast, there was no further change in neglect score after the subsequent session of PA combined with sham tDCS (Time: $F_{(5,25)} = .205$, $p = 0.957$; $\eta^2_p = 0.039$). Rather, the improvement observed after PA + a tDCS was retained across all tests before and after the sham + PA intervention (i.e. for at least 88 days after PA + M1 a-TDCS). Given the longitudinal design, which entails different pre-PA baseline values for the anodal and sham phases, the direct statistical contrast of anodal versus sham was performed on change scores normalized to each phase's pre-intervention baseline mean (ie: post-pre mean percent change

in neglect, for each post-test timepoint, separately for sham and anodal). ANOVA on these percent change in neglect scores with factors of tDCS (2) and Time (5) revealed a significant effect of tDCS (F(1,5)=10.801, p=0.011, 1-tail; $\eta^2_p$ =0.684; mean difference: −16.019, SEM = 4.874; Cohen's d = 1.52). This confirmed the key hypothesis: there was significantly greater improvement in neglect after PA +a tDCS compared to PA + sham. There was no effect of Time (F(4, 20)=1.647, p=0.202, $\eta^2_p$ = 0.248) and no tDCS x Time interaction (F(4, 20) =2.593, p=0.0068, $\eta^2_p$ = 0.341), indicating that the effect was stable across the time period of assessment.

In Phase B (77 days, sessions 5, 6), we again tested whether M1 anodal tDCS would enhance the clinical efficacy of PA. Patient 1 underwent two further PA interventions, with neglect assessed using a new battery of tests, administered twice in the weeks preceding PA, and five times in the subsequent 3 + weeks (**Figure 7B**). During Phase B, the PA procedure involved the Adaptation phase only (no Washout or Retention). The goal of phase B was to further verify the apparent therapeutic effect observed in phase A with anodal tDCS, by aiming to replicate it while ruling out confounding factors. By changing the neglect test battery, we aimed to minimize learning effects. By reversing the order of stimulation (sham, anodal), we aimed to address the possibility that a ceiling effect had precluded further clinical improvement in the Phase A sham condition.

The same statistical analyses as above confirmed that the pattern observed in Phase A (**Figure 7A**) was replicated in Phase B (**Figure 7B**). After PA + sham tDCS, there was again no change in Patient 1's neglect (RM ANOVA on percent neglect scores for 10 sub-tests over Time (6 levels: mean pre-PA versus 5 post-tests): F(5,45)=.423, p=0.73; $\eta^2_p$ = 0.045). By contrast, after PA + M1 anodal tDCS, there was once again a significant improvement in neglect score (Time: F(5,45)=7.368, p<0.001; $\eta^2_p$ = 0.45; planned contrast (mean post – pre): t (9)=-4.928, p<0.001, 1-tail; mean difference: −19.89, SEM: 4.03; Cohen's d: −0.773). As in Phase A, the clinical gain in Phase B was large (20% absolute, 40% relative), stable, and long-lasting, and had not decayed by the latest post-test interval 46 days later. The direct statistical contrast of anodal versus sham on percentage change in neglect data confirmed the key hypothesis: significantly greater improvement in neglect after PA + anodal tDCS compared to sham. ANOVA on percent change in neglect scores with factors of tDCS (2) and Time (5) revealed a large significant effect of tDCS (F(1,9)=17.226, p<0.001, 1 tail; $\eta^2_p$ =0.657; mean difference: −18.085, SEM = 4.357; Cohen's d = 1.697). There was no effect of Time (F(4, 36) =2.104, p=0.101, $\eta^2_p$ = 0.189) and no tDCS x Time interaction (F(4, 36)=1.047, p=0.0397, $\eta^2_p$ = 0.104), indicating that the effect was stable across the time period of assessment.

Finally, statistical analysis on all the data for Patient 1 (ie: AB-BA; phase A and B combined) further confirmed significantly greater improvement in neglect after PA +a tDCS compared to PA +sham across the 7 month intervention period. RM ANOVA on percent change in neglect scores (post-pre mean) across the 16 sub-tests over 5 post-test timepoints, with factors of tDCS (2) and Time (5) found a large significant effect of tDCS (F(1,15)=29.507, p<0.001, 1-tail; $\eta^2_p$ =0.663; mean difference: 17.311; SEM: 3.186; Cohen's d = 1.663). There was no effect of Time (F(4,60)=2.394, p=0.06, $\eta^2_p$ =0.138) and no tDCS x Time interaction (F(4,60)=3.021, p=0.025, $\eta^2_p$ =0.168), indicating that the effect was stable across the time period of assessment.

To further verify the large, long-lasting clinical response observed in Patient 1, two further treatment-resistant neglect patients received PA combined with sham or M1 anodal tDCS, using the identical PA procedure and the same neglect battery used in Phase B. Neglect was assessed before and after two sessions of PA + tDCS, over 2–3 weeks before and 3 weeks after. Patient 2 underwent PA + tDCS in the order anodal-sham, while Patient 3 received the opposite order (**Figure 7C,D**; **Figure 7—figure supplement 1**, **Figure 7—source data 1**).

Patient 2 also showed a significant (30% absolute, 53% relative) improvement in neglect after PA +M1 a-tDCS which was still present at the latest post-test interval, 46 days after PA (**Figure 7—figure supplement 1**, panel A). RM ANOVA on percent neglect scores for each of 8 sub-tests in the anodal phase revealed a significant effect of Time (6 levels: mean pre-PA versus 5 post-tests; F(5,35)=11.096, p<0.001, 1-tail; $\eta^2_p$ = 0.613). The planned contrast (mean post – pre) confirmed a very large significant improvement in neglect after PA +anodal tDCS (t

(7)=-4.520, p<0.001, 1-tail; mean difference: −32.463, SEM: 7.182; Cohen's d: 1.836). By contrast, there was no therapeutic response to PA combined with sham tDCS (F(5,35)=2.871, p=0.077; $\eta^2_p$ = 0.291). The trend was for neglect performance to decrease. The direct contrast of the anodal versus sham effect was also significant, confirming the key hypothesis. RM ANOVA on percent change in neglect scores across the 8 sub-tests and five post-test timepoints found a large significantly greater improvement in neglect after PA +anodal tDCS compared to sham (effect of tDCS: F(1,7)=18.557, p=0.002, 1-tail; $\eta^2_p$ =0.726; mean difference: −41.591, SEM = 9.655; Cohen's d = 0.714). There was no effect of Time (F(4, 28) =2.061 p=0.113, $\eta^2_p$ = 0.227) and no tDCS x Time interaction (F(4, 28)=1.746, p=0.0.168, $\eta^2_p$ = 0.200), indicating that the effect was stable across the time period of assessment.

Patient 3 (**Figure 7—figure supplement 1**, panel B) also showed no improvement in neglect when PA was coupled with sham tDCS (RM ANOVA on percent neglect scores for 10 sub-tests over Time (6 levels: mean pre-PA versus 5 post-tests): F(5,45)=0.358, p=0.874; $\eta^2_p$ = 0.038). By contrast, after PA combined with M1 anodal tDCS, there was a small (~7% absolute, 14% relative), marginally significant improvement in neglect (Time (F(5,45)=1.365, p=0.255; $\eta^2_p$ = 0.132; planned contrast (post – pre): t(9)=1.781, p=0.05, 1-tail; mean difference: 6.519, SEM: 3.66; Cohen's d: 0.286), which was still evident at the latest post-test interval, 21 days later. The direct contrast of the anodal versus sham effect indicated a trend towards a larger effect in the anodal condition, but this difference did not reach significance (RM ANOVA on percent change in neglect scores across 10 sub-tests and five post-test timepoints: tDCS: F (1,9)=1.784, p=0.107, 1-tail; $\eta^2_p$ =0.165; mean difference: 7.416, SEM: 5.55; Cohen's d: 0.434).

Group statistics on Phase B percent change in neglect data combining all 3 patient datasets tested the key prediction of a greater improvement in neglect after PA +anodal tDCS compared to sham at the group level (**Figure 7C**). RM ANOVA on percent change in neglect data across all subtests of the Phase B battery assessed factors of tDCS (2) and Time (5), with Patient (3) as a covariate. There was a large effect of tDCS, indicating significantly greater improvement in neglect with anodal tDCS compared to sham (F(1,26)=7.648, p=0.005, 1-tail; $\eta^2_p$ = 0.227; mean difference: −20.99, SEM = 4.458; Cohen's d = 0.899). There was no effect of Time (F(4, 28)=2.061 p=0.113, $\eta^2_p$ = 0.227) and no tDCS x Time interaction (F(4, 28)=1.746, p=0.0.168, $\eta^2_p$ = 0.200), indicating that this effect was stable across the time period of assessment. The effect of Patient was also significant, reflecting inter-individual variation in the magnitude of this anodal-sham therapeutic difference (F(1,26)=6.427, p=0.018, $\eta^2_p$ =.198).

In summary, both at the level of individual patients and at the group level, in all cases, the combination of PA +M1 a-tDCS caused a significant and long-lasting improvement in neglect. By contrast, when PA was combined with sham tDCS there was no therapeutic response.

## Stimulation-augmented PA therapy for neglect facilitates reaching speed

Given that M1 stimulation induced a sustained leftward shift in reach accuracy without vision (ie. prism after-effect) and in spatial cognition (neglect), we tested for unintended negative motor side effects (i.e. impaired leftward reaching). Detailed kinematic measures of the right arm were recorded while patients performed a naturalistic spatial reaching task, in which they were required to perform a centre-out reaching movement with the ipsilesional (right) arm to grasp a tennis ball, and then subsequently place it in a basket in either the left or right hemifield depending on a central visual instruction pre-cue. Movements were naturally-paced (i.e. no speed instructions). Outlier trials were defined as those in which movement speed (RT and/or MT) exceeded ± 2 SD of the patient's mean for that condition. Outliers were removed prior to analysis. Kinematic parameters for the centre-out phase of the reach (only) were analysed before and after PA + tDCS. All data are in **Supplementary file 2**.

At baseline, we expected the task to be sensitive to neglect, since neglect patients often exhibit 'directional hypokinesia' (i.e. slower movements to the left than to the right). To test this directional hypothesis, kinematic parameters from all trials for each of the three patients' first task session prior to PA/tDCS were analysed (see **Supplementary file 2**). For Patients 1 and 2 these data were for the session before PA + sham tDCS. For Patient 3 these data were for the session before PA + M1 a-tDCS. Across each of the kinematic parameters, RM ANOVA on single trial data analysed the within-subjects factor of Hemifield (left, right) with Patient as a

covariate. This confirmed that patients did indeed exhibit slower movements to left-cued targets compared to right. Three parameters were significant: movement time (MT), peak velocity (PV) and time to peak velocity (TPV): effect of Hemifield on movement time ($F(1,23)$ =7.803, p=0.005, 1-tail; $\eta^2_p$ =0.253; mean difference: 155.2 ms, SEM = 62.984), peak velocity ($F(1,23)$=2.919, p=0.05, 1-tail; $\eta^2_p$ =0.113; mean difference: 26.44 mm/s, SEM = 27.956) and time to peak velocity ($F(1,23)$=4.64, p=0.021, 1-tail; $\eta^2_p$ =0.168; mean difference: 64.8 ms, SEM = 55.707).

Next we tested whether this baseline pattern of hemifield asymmetry was modulated by PA (+sham tDCS), using RM ANOVA to test for an interaction between the within-subjects factors of Hemifield (left, right) and Time (pre/post PA + sham tDCS), with Patient as a covariate. Patient 3 could not be included in this analysis as he did not perform this task during the PA + sham tDCS intervention phase. Analysis was conducted only on the three variables (MT, PV, TPV) that exhibited hemifield asymmetry (L > R) at baseline. There was a significant Hemifield x Time interaction only for MT and TPV (MT: ($F(1,16)$=9.772, p=0.007, 2-tail; $\eta^2_p$ =0.379; TPV: ($F(1,16)$=5.436, p=0.033, 2-tail; $\eta^2_p$ =0.28). PA abolished the baseline (L > R) hemifield asymmetry, reflected in a slowing of both parameters after PA compared to before, specifically on right-cued trials (post-pre: MT = mean difference: 322.778 ms, SEM = 64.988; TPV = mean difference: 121.667 ms, SEM = 61.256) with no change on left-cued trials.

Next we tested whether M1 a-tDCS modulated this effect of PA on these parameters using RM ANOVA with within-subjects factors of Hemifield (left, right), Time (pre/post PA) and tDCS (sham, anodal), with Patient as a covariate. Of the two kinematic variables (MT, TPV) that were both sensitive to neglect at baseline (i.e. L > R) and were modulated by PA (i.e. changed post-pre), only the movement time parameter was further altered by stimulation (i.e. significant interaction of tDCS x Hemifield x Time).

Movement time was speeded after PA + M1 anodal tDCS compared to sham, regardless of hemifield cued. This was reflected in a main effect of tDCS ($F(1,16)$=57.242, p<0.001, $\eta^2_p$ =0.782) which interacted with Time ($F(1,16)$=115.948, p<0.001, 2-tail; $\eta^2_p$ =0.879) and Hemifield (tDCS x Hemifield x Time: $F(1,16)$=9.603, p=0.007, 2-tail; $\eta^2_p$ =0.375). To understand the nature of this tDCS effect, follow-up RM ANOVA on the data for the pre-PA session revealed only a main effect of tDCS, reflecting slower movement times during the baseline anodal session than in the baseline sham session ($F(1,16)$=149.966, p<0.001, 2-tail; $\eta^2_p$ =0.904). Follow-up ANOVA on the post-PA session showed that stimulation reversed this baseline difference, with overall movement times now faster after PA + anodal tDCS compared to sham ($F(1,17)$=11.789, p=0.003, 2-tail; $\eta^2_p$ =0.410). This tDCS effect interacted with Hemifield ($F(1,17)$=7.002, p=0.017, 2-tail; $\eta^2_p$ =0.292), reflecting a larger speeding of movement time for right-cued targets (mean: 173 ms) than for left (mean: 34 ms).

Hence, the combination of PA + M1 a-tDCS (compared to sham) facilitated patients' reaching performance, reflected in speeded movement time both for left- and right-cued target trials. Thus, the sustained leftward shift in pointing and cognitive after-effects induced by stimulation-augmented PA therapy did not have an unintended negative side effect of slowing leftward motor performance. Instead, the beneficial effect for visual neglect was accompanied by a general facilitation of patients' reaching speed, whether pre-cued to the left or to the right.

Since Patient 3 lacked sham session data and so could not be included in the Anodal versus Sham analyses above, MT data from his two anodal sessions were analysed to test whether he also exhibited a similar motor facilitation effect of PA + M1 a-tDCS. For the ipsilateral (right) arm (i.e. arm analysed for Patients 1 and 2), RM ANOVA with factors Hemifield (left, right) and Time (pre, post-PA) confirmed that movement time was speeded significantly after PA + M1 a-tDCS compared to before (Time: $F(1,6)$=6.169, p=0.048, 2-tail; mean difference: 82.857, SEM = 33.361). The absence of an interaction with Hemifield (p=0.8) indicated that, as for patients 1 and 2, movement time was speed both for left-cued and right-cued trials.

For Patient 3's recovered paretic (left) arm, there was no significant change in movement time after PA compared to before (Time: p=0.65). However, the number of successful trials performed with the paretic limb did increase. Prior to PA + M1 a-tDCS, Patient 3 performed only 6/11 (left) and 4/11 (right) correct trials with the paretic limb (*Supplementary file 2*).

However, after PA + M1 a-tDCS, this number increased to 10/11 (left) and 11/11 (right) trials. Whether this accuracy improvement was an effect of PA + M1 a-tDCS or merely an effect of task repetition is unclear.

In summary, the positive effects of PA + M1 a-tDCS for visual neglect were not accompanied by negative side effects on a motor task designed to measure everyday visually-guided reaching behaviour. In fact, stimulation combined with PA caused a generalized improvement in motor performance, reflected in faster reach movement times, regardless of whether a spatial pre-cue instructed subsequent movements to the left or to the right.

