## [Decision Letter]

[Editors’ note: this article was originally rejected after discussions between the reviewers, but the authors were invited to resubmit after an appeal against the decision.]

Thank you for submitting your work entitled "Induced sensorimotor cortex plasticity remediates chronic treatment-resistant visual neglect" for consideration by *eLife*. Your article has been reviewed by three peer reviewers, and the evaluation has been overseen by a Reviewing Editor and myself (Richard Ivry) as the Senior Editor. The following individuals involved in review of your submission have agreed to reveal their identity: Dennis Schutter (Reviewer #1).

Our decision has been reached after consultation between the reviewers. Based on these discussions and the individual reviews below, we regret to inform you that your work will not be considered further for publication in *eLife*. All reviewers felt that the study was conceptually innovative and exciting in its novelty. However, all reviewers found consistent methodological problems that may not be solved in a revised version.

Reviewer #1:

The paper describes a proof-of-concept study that combines prism adaption with anodal DC stimulation to enhance consolidation of sensorimotor and cognitive effect of prism effects. These findings may have a large impact on the treatment of chronic visual neglect.

This is an exciting study that is both highly original and timely which may prove to be the first crucial step for finding new ways to improve chronic visual neglect in stroke patients.

The data look very promising, but despite my enthusiasm for this study by renowned experts, the results are in my view too preliminary to the published in *eLife* at this time. First and most importantly, the experiments are susceptible to Type I and Type II errors. Secondly, the experiment that deals with the critical part of the study consists of only three patients without metabolic indices to compare to the healthies.

Related to the previous point, the direct comparison between healthy volunteers and chronic stroke patients concerning the working mechanisms is far from straightforward. For example, simply the age difference between healthies and patients is a serious concern when it comes to neuroplasticity.

Furthermore, there is no evidence for any mechanism that can explain these effects or challenge current ideas. In fact, the effects observed for anodal DC stimulation are exactly in line with the Bienenstock Munro Cooper theory on synaptic modification of cortical hyperexcitability.

Also, it is not clear why the authors did not use paired-pulse TMS to measure intracortical inhibition to obtain a more straightforward measure which can be more readily linked to the well documented literature on the effect of DC stimulation on M1 excitability.

In sum, this study holds great promise as this could be the start of a very promising and important line of a novel research-oriented development treatment, but in its present form I think is more suitable for a clinically oriented (neurology) journal.

Reviewer #2:

In the present manuscript by Jacinta O'Shea and colleagues entitled "Induced sensorimotor cortex plasticity remediates chronic treatment-resistant visual neglect", the authors showed that anodal tDCS applied during prism adaptation over the left motor cortex caused a lasting improvement of neglect symptoms. The paper deals with an interesting and up-to-date issue in the neurorehabilitation field. Especially, the two-step bench-to-bedside translational approach, firstly demonstrating the effect of the combined treatment in healthy volunteers and then assessing its efficacy in neglect patients, represents a strength of this challenging study.

– One strength of the present work relies on the investigation of the causal mechanisms that induced AE enhancement consolidation by M1 tDCS. The investigation of the relation between the behavioral and the neurochemical effect of M1 a-tDCS by means of MRS, further enriches the relevance of this manuscript but correlation is not the correct way to investigate a causal mechanism. In fact, correlation does not imply causality and concluding that the observed behavioral effect depends causally on a reduction of M1 cortical inhibitory tone is a fallacy. Equally, sentences as "Our MRS data suggest that pathological disinhibition of M1 is necessary for stimulation-enhanced PA consolidation to occur" (Discussion section) or "a magnetic resonance spectroscopy experiment was conducted as a physiological test of the hypothesis that the effect predicted in part 1 would depend causally on disinhibition (GABA decrease) in M1" (Materials and methods section) are not correct and need to be consistently modified.

– Some of the finding presented in the Results section of the experimental phase are not entirely supported by the actual results of statistical analysis reported in the supplementary material. The key effects summarized in the Results section are inferred by planned contrasts not allowed by the main effect which turned out to be not significant. Specifically:

– "M1 a-tDCS increased AE magnitude during Adaptation": for the group that had stimulation during adaptation the main effect of tDCS was not significant (p=0.191), although planned contrasts against sham revealed a tendency for anodal stimulation to increase AE magnitude during Adaptation (1-tail p=0.042) whereas cathodal had no effect (p>0.6).

– "M1 a-tDCS during Adaptation enhanced subsequent AE persistence during Washout and Retention": for the group that had stimulation during adaptation, although the main effect of tDCS was not significant (p=0.093), planned contrasts revealed significantly greater AE retention selectively with M1 anodal tDCS compared to sham (1-tail p=0.034).

– Some recent literature relevant for the current work is missing and needs to be reported and deeply discussed. Indeed, Calzolari and colleagues (Calzolari et al., 2015) applied a-tDCS over the left PPC and over the left cerebellum reporting that the stimulation restored the leftward AE and abolished the rightward deviation, respectively. Furthermore, the very recent results showed by Panico and colleagues (Panico et al., 2017) and by Bracco and coworkers (Bracco et al., 2017) need to be integrated in the manuscript. Another issue that is not considered in the Introduction but deserves further attention regards to the explanation of the control sites adopted (left parietal cortex and right cerebellum). As the posterior parietal cortex and the cerebellum are two key cortical sites of a bilateral network involved in PA, the authors should integrate this topic.

When discussing the results, the authors should consider also several studies which, differently from conventional models, found that neglect symptomatology and its recovery does not depend on the inhibition of the left undamaged hemisphere (see for example Ricci et al., 2012; Bagattini et al., 2014; Cappa and Perani, 2010; Thimm et al., 2008).

– Considering the complexity and the ambition of the present study, I think the manuscript would benefit from a clearer description and organization of the study. I would suggest streamlining the text, making it clearer and less overloaded, especially for the Results and Method section. In my opinion, the statistical analysis results reported in the supplementary material is very clear and understandable so I would suggest to add it in the main text or at least to take the cue from it. Again, I would suggest indicating in the Result section and in the figures the experiments to which the authors refer.

Reviewer #3:

The authors examined the effect of M1-a tDCS on the retention of the aftereffect (AE) of prism adaptation (PA) in healthy volunteers and four patients with chronic neglect. They find that this stimulation (but not other stimulation protocols) improves retention for PA AEs, that stimulation effects depend on reduction of GABA concentration in M1 and that the treatment benefit of PA for neglect patients is prolonged by M1-a tDCS stimulation. It is concluded that combined M1-a tDCS stimulation during PA might induce long lasting treatment benefits in neglect patients. Interestingly, left-hemispheric tDCS can improve performance in neglect patients. These findings seem to be in conflict with the claim that an overactive left hemisphere is responsible for the neglect symptoms.

This study has significant relevance and potential. Overall it is my impression that this study contains the promise of not just one but several interesting papers. The two most relevant issues addressed by this paper relate to the role of M1-tDCS for AE retention in healthy volunteers and the potential of M1-tDCS to turn a single-session intervention into a neglect-treatment with long-lasting benefits. However, I am not convinced that the presented evidence is sufficient to support the paper's conclusions.

In recent years the reliability of effects previously reported for tDCS have been questioned. Horvath and others argued that tDCS are either non-existent or smaller than previously assumed. It is suggested that reliable estimates for tDCS effects can only be expected with moderate to large sample sizes. In light of these comments, the sample sizes used in this study seem quite small. Nine healthy volunteers were enrolled for each of the individual experiments, and four single-case studies were used to demonstrate the benefit of tDCS on neglect patients. Neither healthy volunteers nor neglect patients are in such short supply that larger sample sizes seem impossible. The small samples in conjunction with some other methodological concerns (see below) lead to the conclusion that it might be best to await further data collection before we evaluate the validity of the proposed claims.

Specific issues:

1) Post-hoc comparisons using α-adjustments for multiple tests are avoided; instead one-tailed planned comparisons are carried out. To me this seems only acceptable if:

a) The predictions are convincingly justified.

b) The planned comparisons are limited to a small set of pre-determined conditions and variables.

The main prediction (i.e. that specifically M1-a-tDCS) stimulation will enhance AE retention is hardly justified at all. In fact, given the known involvement of parietal and cerebellar structures in PA, stimulation over cerebellar or parietal sites might have been expected to be just as effective. Moreover, for many questions several time-points and several variables are subjected to planned comparisons. In this case multiple comparison corrections seem called for even if the direction of the expected effect may have been expected.

2) In the experiment with healthy volunteers looking at retention over a time-course of 5 days a within-subject design is used. Both anodal and sham conditions are examined in the same set of participants in different weeks. Assuming that PA will have long-lasting effects carry-over effects from sham to anodal and vice versa may be expected. Consequently, it is hard to tell which of the effects observed in the sham or anodal phase are due to the sham-PA intervention and which due to the anodal-PA intervention.

3. Neglect patients. All three neglect patients suffer both from neglect and hemianopia. Given that neglect and hemianopia are separate clinical entities associated with different types of brain damage and different etiologies this coincidence is quite surprising. Does this mean that we have a very unusual patient sample? Is it possible that severe neglect may have been misdiagnosed as hemianopia? There is hardly any information on the hemianopia and it seems the issue of hemianopia has been only addressed at the beginning. It would be interesting to learn whether the hemianopia also benefited from the treatment.

4) To assess the clinical benefits neglect patients were repeatedly tested with the same paper and pencil tests. This is problematic. The test batteries do not provide a sufficient number of parallel test versions. Practice effects can be expected and may masquerade as treatment effects.

5) Neglect case series: AE retention is tested in patient 1 but not in patients 2-4. This is a shame it would be interesting to see whether AE retention correlates with the retention of clinical benefits. The group analysis on patients 2-4 seems problematic. I am not entirely sure but it seems that the analysis is carried out at the level of individual blocks and that blocks from one patient are mixed together with blocks from the other patients. I suspect that this procedure violates the assumption of statistical independence.

6. Naturalistic spatial reaching task. I could not find the results and analysis based on the findings from this task.

7. Supplementary material: Section 7. For the ANOVA, effects from three sham sessions are combined and compared to the findings from one single anodal session. This seems problematic given that the sham sessions contained three times more trials than the anodal session.

8. Finally a theoretical question: does the fact that M1-tDCS over the left hemisphere improved neglect signs really contradict the Kinsbourne/Corbetta hypothesis that hemispheric asymmetry is at the heart of the attentional deficit found in neglect patients? We know very little about the precise effects of tDCS on the brain – possibly not enough to exclude the possibility that left M1-tDCS may also lead to some reduction in left-hemispheric activation.

Conclusion: My review sounds more critical that it is intended to be. I like the study and I think the study has enormous potential. In my view it would have been more convincing to focus on just one of the three questions and address this question in a between-subject design using a moderately sized sample of healthy volunteers (question 1) or neglect patients (question 2).

[Editors’ note: what now follows is the decision letter after the authors submitted for further consideration.]

After extended discussion between myself and the Reviewing Editor, we are convinced that the arguments you raise in your appeal have merit and are reversing our rejection decision. We invite you to submit a revised version of your manuscript.

This puts us in an unusual situation, at least with respect to the *eLife* approach. Typically, revise and resubmit comes with an integrated decision letter that synthesizes the reviews and makes specific requests. You have all of the reviews from the rejection and I am confident you will give careful consideration to all of the comments. Nonetheless, I suspect you would appreciate some guidance on how to respond to them. To this end, I have provided annotation below (using my initials RI) on the major comments of each of the three reviewers. I also ask that you provide a clearer justification/motivation for the sites you've chosen for stimulation (experimental and control), and note spatial limitations with tDCS in localizing effects to particular brain regions.

Annotated comments on the major comments of the three reviewers:

Reviewer #1:

The data look very promising, but despite my enthusiasm for this study by renowned experts, the results are in my view too preliminary to the published in *eLife* at this time. First and most importantly, the experiments are susceptible to Type I and Type II errors.

RI: Some sort of power/sensitivity analysis would address concerns about Type I and Type II errors. If you do find the work underpowered (initial control study), please increase the sample size to an appropriate level (and I recognize it isn't completely kosher to add participants in this way but think that given you aren't doing this to "reach" magical.5, it's ok).

Secondly, the experiment that deals with the critical part of the study consists of only three patients without metabolic indices to compare to the healthies.

RI: You can certainly acknowledge that the sample is small, but I don't think this is critical concern given you've taken a case-study approach.

Related to the previous point, the direct comparison between healthy volunteers and chronic stroke patients concerning the working mechanisms is far from straightforward. For example, simply the age difference between healthies and patients is a serious concern when it comes to neuroplasticity.

RI: Please review carefully to make clear that your main points do not involve comparisons between controls and patients, but rather the patient study is self-contained with sham providing the control.

Also, it is not clear why the authors did not use paired-pulse TMS to measure intracortical inhibition to obtain a more straightforward measure which can be more readily linked to the well documented literature on the effect of DC stimulation on M1 excitability.

RI: You might comment on this in the Discussion as an approach that could be used in the future to explore the underlying mechanisms.

Reviewer #2:

– One strength of the present work relies on the investigation of the causal mechanisms that induced AE enhancement consolidation by M1 tDCS. The investigation of the relation between the behavioral and the neurochemical effect of M1 a-tDCS by means of MRS, further enriches the relevance of this manuscript but correlation is not the correct way to investigate a causal mechanism. In fact, correlation does not imply causality and concluding that the observed behavioral effect depends causally on a reduction of M1 cortical inhibitory tone is a fallacy. Equally, sentences as "Our MRS data suggest that pathological disinhibition of M1 is necessary for stimulation-enhanced PA consolidation to occur" (Discussion section) or "a magnetic resonance spectroscopy experiment was conducted as a physiological test of the hypothesis that the effect predicted in part 1 would depend causally on disinhibition (GABA decrease) in M1" (Materials and methods section) are not correct and need to be consistently modified.

RI: Although it is important to recognize the limitations of correlational work, you can certainly highlight how the GABA part contributes to possible mechanisms for the observed effects.

– Some of the finding presented in the Results section of the experimental phase are not entirely supported by the actual results of statistical analysis reported in the supplementary material The key effects summarized in the results are inferred by planned contrasts not allowed by the main effect which turned out to be not significant. Specifically:

– "M1 a-tDCS increased AE magnitude during Adaptation": for the group that had stimulation during adaptation the main effect of tDCS was not significant (p=0.191), although planned contrasts against sham revealed a tendency for anodal stimulation to increase AE magnitude during Adaptation (1-tail p=0.042) whereas cathodal had no effect (p>0.6).

– "M1 a-tDCS during Adaptation enhanced subsequent AE persistence during Washout and Retention": for the group that had stimulation during adaptation, although the main effect of tDCS was not significant (p=0.093), planned contrasts revealed significantly greater AE retention selectively with M1 anodal tDCS compared to sham (1-tail p=0.034).

RI: Please review these issues and modify as appropriate.

– Some recent literature relevant for the current work is missing and needs to be reported and deeply discussed…

RI: Please consider the relevance of the various papers cited by the reviewer.

– Considering the complexity and the ambition of the present study, I think the manuscript would benefit from a clearer description and organization of the study. I would suggest streamlining the text, making it clearer and less overloaded, especially for the Results and Method section. In my opinion, the statistical analysis results reported in the supplementary material is very clear and understandable so I would suggest to add it in the main text or at least to take the cue from it. Again, I would suggest indicating in the Result section and in the figures the experiments to which the authors refer.

RI: I will defer to you on whether to integrate more of the supplemental information in the main manuscript. I found the flow of the paper to be reasonable, but do look for ways to add signposts as you move from one section to the next.

Reviewer #3:

In recent years the reliability of effects previously reported for tDCS have been questioned. Horvath and others argued that tDCS are either non-existent or smaller than previously assumed. It is suggested that reliable estimates for tDCS effects can only be expected with moderate to large sample sizes. In light of these comments, the sample sizes used in this study seem quite small…

RI: As noted above, an assessment of the power/sensitivity would be helpful here. I am familiar with the Horvath paper (and the responses to it). Certainly it’s important to note that there are concerns with reliability of tDCS.

1) Post-hoc comparisons using α-adjustments for multiple tests are avoided; instead one-tailed planned comparisons are carried out. To me this seems only acceptable if:

a) The predictions are convincingly justified.b) The planned comparisons are limited to a small set of pre-determined conditions and variables.

RI: Per above, please review your selection and justification for the statistical analyses.

3) Neglect patients. All three neglect patients suffer both from neglect and hemianopia. Given that neglect and hemianopia are separate clinical entities associated with different types of brain damage and different etiologies this coincidence is quite surprising. Does this mean that we have a very unusual patient sample? Is it possible that severe neglect may have been misdiagnosed as hemianopia? There is hardly any information on the hemianopia and it seems the issue of hemianopia has been only addressed at the beginning. It would be interesting to learn whether the hemianopia also benefited from the treatment.

RI: It would be useful to comment on the hemianopia issue and in particular, if your sample is atypical. If you have information on pre-post hemianopia testing, this could be interesting since I'm betting the hemianopia did not change (thus providing another argument for specificity of the improvement).

4) To assess the clinical benefits neglect patients were repeatedly tested with the same paper and pencil tests. This is problematic. The test batteries do not provide a sufficient number of parallel test versions. Practice effects can be expected and may masquerade as treatment effects.

RI: You could acknowledge this limitation.

5) Neglect case series: AE retention is tested in patient 1 but not in patients 2-4. This is a shame it would be interesting to see whether AE retention correlates with the retention of clinical benefits. The group analysis on patients 2-4 seems problematic. I am not entirely sure but it seems that the analysis is carried out at the level of individual blocks and that blocks from one patient are mixed together with blocks from the other patients. I suspect that this procedure violates the assumption of statistical independence.

RI: Per above in terms of considering your statistics.

6) Naturalistic spatial reaching task. I could not find the results and analysis based on the findings from this task.

RI: Please clarify.

7) Supplementary material: Section 7. For the ANOVA, effects from three sham sessions are combined and compared to the findings from one single anodal session. This seems problematic given that the sham sessions contained three times more trials than the anodal session.

RI: Per above in terms of considering your statistics.

8) Finally a theoretical question: does the fact that M1-tDCS over the left hemisphere improved neglect signs really contradict the Kinsbourne/Corbetta hypothesis that hemispheric asymmetry is at the heart of the attentional deficit found in neglect patients? We know very little about the precise effects of tDCS on the brain – possibly not enough to exclude the possibility that left M1-tDCS may also lead to some reduction in left-hemispheric activation.

RI: This is an interesting point to consider in terms of placing your work within current theoretical accounts of neglect.

---

## [Author Response]

[Editors’ note: the author responses to the first round of peer review follow.]

*Thank you for submitting your work entitled "Induced sensorimotor cortex plasticity remediates chronic treatment-resistant visual neglect" for consideration by eLife. Your article has been reviewed by three peer reviewers, and the evaluation has been overseen by a Reviewing Editor and myself (Richard Ivry) as the Senior Editor. The following individuals involved in review of your submission have agreed to reveal their identity: Dennis Schutter (Reviewer #1).*

*Our decision has been reached after consultation between the reviewers. Based on these discussions and the individual reviews below, we regret to inform you that your work will not be considered further for publication in eLife. All reviewers felt that the study was conceptually innovative and exciting in its novelty. However, all reviewers found consistent methodological problems that may not be solved in a revised version. […]*

The reviewers raised methodological concerns, which are cited as the key reason for rejection of our paper. We can see that the way these concerns are presented does indeed give the (misleading) impression that there are serious problems with our work. However, the methodological concerns raised are not in fact present in our manuscript. Hence, we are concerned that the decision to reject our paper was based on incorrect information.

The reviewers raised 15 major methodological concerns. Of these, 3 are logically incorrect (R2 point 1, R3 point 1, 3), 6 are factually incorrect (R2 point 2, R3 point 2, 4, 5, 6, 8), and 6 are irrelevant (R1 points 1-5; R2 point 3).

For the sake of brevity, we do not itemize all here. However, to give one example, reviewers 2 and 3 questioned our statistical analysis. Our manuscript reports a body of hypothesis driven experiments that repeatedly test a single a priori defined directional prediction: that M1 anodal tDCS during prism adaptation will increase retention of the visuomotor and cognitive after-effects. To test this statistically, we pre-specified a single planned contrast (Anodal>Sham, i.e. 1-tail), which we applied consistently throughout every experiment in the manuscript. As the editors will understand quite straightforwardly, this is an appropriate and entirely standard way to conduct hypothesis-driven science, and to test an a priori-defined one-way directional prediction.

Reviewer 2 (point 2) claims that planned contrasts are "not allowed" unless higher order ANOVA main effects are significant. This is incorrect. Reviewer 3 (point 2) specifies (correctly) the requirements for conducting planned contrasts (which our manuscript fulfils). However, s/he then claims: " for many questions several time-points and several variables are subjected to planned comparisons. In this case multiple comparison corrections seem called for". This is incorrect. We do not do this. As described above, we test a single prediction (not "many questions") about a single variable, the prism after-effect (not "several variables") during a single time-point, the post-adaptation period (not "several time-points"). We do so using a single planned contrast (Anodal>Sham), so "multiple comparison correction" does not arise. Reviewer 3 makes additional factually incorrect claims about our statistics (points 6, 8).

We therefore request the opportunity to provide a detailed rebuttal, explaining how each and every point is incorrect or irrelevant.

[Editors’ note: the author responses to the re-review follow.]

*This puts us in an unusual situation, at least with respect to the eLife approach. Typically, revise and resubmit comes with an integrated decision letter that synthesizes the reviews and makes specific requests. You have all of the reviews from the rejection and I am confident you will give careful consideration to all of the comments. Nonetheless, I suspect you would appreciate some guidance on how to respond to them. To this end, I have provided annotation below (using my initials RI) on the major comments of each of the three reviewers. I also ask that you provide a clearer justification/motivation for the sites you've chosen for stimulation (experimental and control), and note spatial limitations with tDCS in localizing effects to particular brain regions.*

We have expanded the Introduction to better clarify our rationale for targeting M1 (experimental site). We now also provide a clearer justification for our choice of anatomical control sites, PPC and cerebellum, in the main text of the Results. In the revised Discussion we address spatial limitations in localizing effects of tDCS to particular brain regions, and in so doing, we also highlight how our MRS experiment (Figure 4) provides independent physiological evidence to ground our M1 localization claim.

Revised text (Introduction):

“We targeted M1 based on evidence that this brain region mediates the early consolidation of motor memories formed during adaptation (Hadipour-Niktarash, Lee, Desmond, & Shadmehr, 2007; Hunter, Sacco, Nitsche, & Turner, 2009; Landi, Baguear, & Della-Maggiore, 2011; Li, Padoa-Schioppa, & Bizzi, 2001; Richardson et al., 2006). […] Hence, by tonically disinhibiting M1 via a-tDCS during adaptation, we aimed to potentiate the earliest memory traces formed during adaptation.”

Results, “Experimental Phase” section:

“Figure 2 summarizes the AE results for this key experiment and for several control experiments. […] Lesions to parietal cortex (Newport, Brown, Husain, Mort, & Jackson, 2006; Pisella et al., 2004) and cerebellum (Martin, Keating, Goodkin, Bastian, & Thach, 1996; Weiner, Hallett, & Funkenstein, 1983) have been shown to disrupt the magnitude of error correction and prism after-effects, but have not been shown to specifically affect consolidation.

Discussion, sixth paragraph:

“The tDCS protocol used here is routinely applied in motor cortex studies (1mA, 20 minutes, 7 x 5 cm electrodes, anode over left M1, cathode over right eyebrow). […] More specifically, the present data suggest that this physiology may causally constrain the impact of motor memory plasticity induction via M1 a-tDCS.”

*Reviewer #1:*

*The data look very promising, but despite my enthusiasm for this study by renowned experts, the results are in my view too preliminary to the published in eLife at this time. First and most importantly, the experiments are susceptible to Type I and Type II errors.*

*RI: Some sort of power/sensitivity analysis would address concerns about Type I and Type II errors. If you do find the work underpowered (initial control study), please increase the sample size to an appropriate level (and I recognize it isn't completely kosher to add participants in this way but think that given you aren't doing this to "reach" magical.5, it's ok).*

As stated previously (Materials and methods, Study Design), we estimated the required sample size (n=9) for the healthy volunteer experiments based on a prior study that used a related paradigm (M1 a-tDCS during adaptation to visuomotor rotation) (Galea et al., 2010). That study used a between-subjects sample of n=10, and found a significant increase in retention with anodal tDCS compared to sham, using 2-tailed hypothesis testing. Hence, we expected that our within-subjects design, testing a 1-tailed hypothesis (Anodal>Sham), with a similar sample size, would be more sensitive, and so should have sufficient power. Formal power and sensitivity analyses confirmed this. We calculated the a prioriminimum sample size required for our design, based on the effect size reported in Galea et al., 2010. We also conducted sensitivity analysis on our design a posteriori. Both analyses confirmed we had sufficient power/sensitivity and hence that Type 1 and 2 error rates were controlled appropriately.

Revised text (Methods, section “Study Design”):

“An a priori sample size calculation, based on the effect size reported in Galea et al. 2011, confirmed this (for 1-tail t-test with effect size dz = 0.92, β = 80%, α = 0.5, required n = 9). A posteriori sensitivity analysis based on our observed results (sensitivity of 1-tail t-test with n=9, β = 80%, α = 0.5, effect size dz = 0.9) further confirmed that our design and analysis strategy controlled Type 1 and 2 error rates appropriately. These statistical analyses were performed using G* power software (Faul, Erdfelder, Lang, & Buchner, 2007).”

On the view that our results are "too preliminary" we respectfully disagree. The key criterion for assessing any study is whether the evidence is sufficient to support the conclusions given the study goal. Our goal was to test the hypothesis that M1 a-tDCS would enhance consolidation of prism adaptation, causing: 1) a sustained leftward pointing after-effect, and 2) lasting improvement in visual neglect. Prediction 1 was confirmed (Figure 2) and then replicated in two independent experiments (Figure 3, Figure 4 source file 2). Prediction 2 was confirmed in a longitudinal single case study in Patient 1 (Figure 7), and then replicated within-subject (Figure 7). It was then replicated between-subjects in two additional longitudinal case series, in Patient 2 and Patient 3 (Figure 7, Figure 7—figure supplement 1). Thus, our 2 key predictions were confirmed, and replicated, constituting sufficient evidence to establish scientific proof of concept.

*Secondly, the experiment that deals with the critical part of the study consists of only three patients without metabolic indices to compare to the healthies.*

*RI: You can certainly acknowledge that the sample is small, but I don't think this is critical concern given you've taken a case-study approach.*

*Related to the previous point, the direct comparison between healthy volunteers and chronic stroke patients concerning the working mechanisms is far from straightforward. For example, simply the age difference between healthies and patients is a serious concern when it comes to neuroplasticity.*

*RI: Please review carefully to make clear that your main points do not involve comparisons between controls and patients, but rather the patient study is self-contained with sham providing the control.*

There are important misunderstandings here.

First, we do not make any direct comparison between the healthy volunteers and the patients anywhere in our paper. This is reflected in the Results section structure, which is divided into two separate phases – experimental (volunteers) and clinical (patients). In each of the patient studies we contrast the effect of Anodal versus Sham tDCS, and each patient acts as his own internal control. We agree that any direct comparison between patients and young healthy volunteers would be problematic for many reasons. Since we do not do so, this concern does not arise.

Second, the reviewer is correct that in the clinical phase in patients we did not acquire "metabolic indices to compare to the healthies". The goal of the patient study was to test a behavioral hypothesis: that tDCS would enhance prism therapy for neglect. Such data would be: 1) irrelevant to this goal; 2) uninterpretable (as agreed above).

Third, the phrase "only three patients" suggests an important misunderstanding about the nature of the experimental design used in the clinical phase. We did not design or conduct a small group (n=3) trial. We designed and conducted a series of longitudinal single case intervention studies. Single case experimental design is a recognised valuable approach for testing the efficacy of therapeutic interventions in the behavioral sciences. In each individual patient, we tracked neglect over several months, and contrasted the effect of prism therapy + real versus sham tDCS within-subject in each case.We confirmed our a priori prediction (Anodal>Sham) in Patient 1 (Figure 7). We then replicated it within-subject in the same individual (Figure 7). Next, we replicated it between-subjects in Patient 2 (Figure 7, Figure 7—figure supplement 1). And then we replicated it again between-subjects in Patient 3 (Figure 7, Figure 7—figure supplement 1). Our results therefore consist of data-rich single case evidence across four independent longitudinal timeseries, in which each patient acts as his/her own control.

*Also, it is not clear why the authors did not use paired-pulse TMS to measure intracortical inhibition to obtain a more straightforward measure which can be more readily linked to the well documented literature on the effect of DC stimulation on M1 excitability.*

*RI: You might comment on this in the Discussion as an approach that could be used in the future to explore the underlying mechanisms.*

Paired-pulse TMS measures the amplitude of motor-evoked potentials in hand muscle. Changes in MEP amplitude with paired-pulse TMS may be interpreted in terms of cortical GABA *by inference*, by referring to previous studies that have combined MEPs with GABAergic drugs. It is unclear why the reviewer regards this as "more straightforward" than MRS, which estimates GABA concentration within the motor cortex directly.

Neither MRS nor paired-pulse TMS provide simple measures of GABA, and the relationship between the two is complex (Stagg et al., 2011). We chose to use MRS for several reasons. There is an established literature combining brain stimulation with MRS. This has shown repeatedly that M1 a-tDCS decreases local GABA concentration (Stagg 2009; Bachtiar 2015; Antonenko 2017), and that MRS-assessed GABA captures meaningful physiological variation that relates quantitatively to individual differences in motor function, learning, and memory stabilization (Stagg 2012, Kim 2014, Barron 2016). For these reasons, we chose MRS to test our hypothesis that the stimulation-induced M1 GABA change would relate to the stimulation-induced M1-dependent functional change (AE retention).

Revised text (Results):

“We hypothesized that a known effect of M1 a-tDCS, transient reduction of M1 cortical inhibitory tone, measured as a reduction in gamma-aminobutyric acid (GABA) concentration (Stagg et al., 2009), may partly mediate the behavioural effect.”

*Reviewer #2:*

*– One strength of the present work relies on the investigation of the causal mechanisms that induced AE enhancement consolidation by M1 tDCS. The investigation of the relation between the behavioral and the neurochemical effect of M1 a-tDCS by means of MRS, further enriches the relevance of this manuscript but correlation is not the correct way to investigate a causal mechanism. In fact, correlation does not imply causality and concluding that the observed behavioral effect depends causally on a reduction of M1 cortical inhibitory tone is a fallacy. Equally, sentences as "Our MRS data suggest that pathological disinhibition of M1 is necessary for stimulation-enhanced PA consolidation to occur" (Discussion section) or "a magnetic resonance spectroscopy experiment was conducted as a physiological test of the hypothesis that the effect predicted in part 1 would depend causally on disinhibition (GABA decrease) in M1" (Materials and methods section) are not correct and need to be consistently modified.*

*RI: Although it is important to recognize the limitations of correlational work, you can certainly highlight how the GABA part contributes to possible mechanisms for the observed effects.*

Our MRS experiment tested for correlation between the magnitude of the neurochemical and behavioral tDCS interference effects, as a way to test a candidate physiological mechanism (GABA decrease) that may mediate the behavioral effect of tDCS (enhanced retention). We agree that there are of course limitations to correlational inference. For instance, we cannot rule out that some unknown third variable could influence both the tDCS-induced GABA change (at rest) and the tDCS-induced behavior change during adaptation, thus accounting for their covariation. We have modified the text to acknowledge this caveat. In the revised text we now also restate our question in terms of physiological sensitivity to tDCS, to further clarify the logic of inferring causal constraints from neurochemistry to behavior.

Revised text (Results):

“By what causal mechanism did M1 stimulation enhance AE consolidation? […] Of course it cannot be ruled out that some unknown third variable influences both the tDCS-induced GABA change (at rest) and the tDCS-induced behaviour change (during adaptation), thus accounting for their covariation.”

*– Some of the finding presented in the Results section of the experimental phase are not entirely supported by the actual results of statistical analysis reported in the supplementary material. The key effects summarized in the results are inferred by planned contrasts not allowed by the main effect which turned out to be not significant. Specifically:*

*– "M1 a-tDCS increased AE magnitude during Adaptation": for the group that had stimulation during adaptation the main effect of tDCS was not significant (p=0.191), although planned contrasts against sham revealed a tendency for anodal stimulation to increase AE magnitude during Adaptation (1-tail p=0.042) whereas cathodal had no effect (p>0.6).*

*– "M1 a-tDCS during Adaptation enhanced subsequent AE persistence during Washout and Retention": for the group that had stimulation during adaptation, although the main effect of tDCS was not significant (p=0.093), planned contrasts revealed significantly greater AE retention selectively with M1 anodal tDCS compared to sham (1-tail p=0.034).*

*RI: Please review these issues and modify as appropriate.*

The reviewer claims that "planned contrasts [are] not allowed…[if] the main effect…turn[s] out to be not significant". This is incorrect. Unlike post hoc analyses, planned contrasts are specified *in advance*, at the same time, and within the same RM ANOVA statistical model, as tests for main effects and interactions. By restricting the number of planned contrasts that can be run, the overall family-wise error rate is controlled. The main effect and the planned contrast ask different questions: the main effect of tDCS in our RM ANOVA asks: "does at least 1 of the conditions differ from at least one other?", encompassing 6 pairwise comparisons (Anodal>/<Sham, Anodal>/<Cathodal, Cathodal>/<Sham). By contrast, the 1-tailed 'simple' planned contrast asks a more specific question: "is the Anodal effect significantly greater than control (Sham)?". It is possible for the planned contrast to be significant and the main effect not (as in our study), since variance across (uninformative) pairwise contrasts (e.g. Anodal versus Cathodal) can reduce the sensitivity of the main effect analysis.

The goal of our study was to test a single *a priori* specified 1-tailed prediction: that M1 a-tDCS during adaptation would enhance retention of the prism after-effect (i.e. anodal>sham). Thus, the planned contrast analysis we used is both valid and the most appropriate/sensitive way to test this hypothesis. In short, our results are indeed supported by our statistical analyses. Our justification for adopting this statistical analysis approach is stated clearly in the Materials and methods, Statistical Analysis section. We have now revised the text to make explicit that planned contrasts are specified at the same time as the main effect and interaction analyses, and are run within the same overall RM ANOVA model.

Revised text (Materials and methods, section “Statistical Analysis”):

“The 1-tailed planned contrast (Anodal>Sham) was specified in advance within the same RM ANOVA model in which main effects and interactions were tested, using the 'simple' planned contrast procedure in SPSS, which contrasts an experimental condition against a control.”

– Some recent literature relevant for the current work is missing and needs to be reported and deeply discussed.

*RI: Please consider the relevance of the various papers cited by the reviewer.*

We have now included the citation to Panico et al., 2017 in the Introduction and the Discussion. The other papers are not relevant to the goal of our study.

Here we explain why each of the other papers is not relevant:

1) Calzolari et al., tests the effect of parietal tDCS in a patient with a cerebellar lesion. The after-effect measure that is the focus of our paper is intact in their patient and it is unaffected by their stimulation.

2) In Bracco et al., healthy controls undergo prism adaptation and tDCS of left M1, each applied sequentially (not concurrently, as in our paper – we found offline stimulation to be ineffective). Even though their participants adapted with the right hand and left M1 was stimulated, their paper does not test for any physiological effect in that modulated circuit (i.e. left motor cortex, which undergoes stimulation and which controls the adapting hand). Instead, it reports only increases in right motor cortex excitability (i.e. M1 controlling the passive left hand which did not adapt and was not stimulated). The entire logic of our study is about tDCS-induced consolidation of motor memory in left sensorimotor circuits controlling the adapting right hand. Since Bracco et al., do not present any data that speak to this issue, it is difficult to see how that study could inform the interpretation of ours.

3) The reviewer cites four papers that s/he claims "found that neglect symptomatology and its recovery does not depend on inhibition of the left undamaged hemisphere". The references do not support this claim. Both Ricci et al., 2012 and Bagattini et al., 2014 studied healthy controls. Cappa & Perani 2010 is a review of early brain imaging studies (mostly PET) of neglect in the 1990s (i.e. no primary research data). Apart from the fact that imaging data are correlative, logically, one cannot (nor do the authors claim to) prove the negative. Thimm et al., 2008 report bilateral increases in brain activity with recovery of neglect over time, which they suggest might be 'compensatory'. However, they also report "increase in right hemisphere and decrease in left hemisphere" activation, which they interpret explicitly as supportive of Corbetta's 2005 'push-pull' model of interhemispheric balance.

*– Considering the complexity and the ambition of the present study, I think the manuscript would benefit from a clearer description and organization of the study. I would suggest streamlining the text, making it clearer and less overloaded, especially for the Results and Method section. In my opinion, the statistical analysis results reported in the supplementary material is very clear and understandable so I would suggest to add it in the main text or at least to take the cue from it. Again, I would suggest indicating in the Result section and in the figures the experiments to which the authors refer.*

*RI: I will defer to you on whether to integrate more of the supplemental information in the main manuscript. I found the flow of the paper to be reasonable, but do look for ways to add signposts as you move from one section to the next.*

Our manuscript aims to strike an appropriate balance between narrative flow and technical detail. We present a concise summary of key findings at the start of the Results section, and then elaborate on each individual finding as the Results narrative progresses. We provide full details of all statistical analyses and additional methods information in the Appendix. We are pleased the reviewer judges this information to be clear and understandable. We do not think it sensible to move the detailed information in the Appendix into the main text of the paper, as readers are then likely to lose the thread of the argument. This would also make it more "overloaded". The consensus of colleagues who have read different versions of our manuscript is that this approach we have taken works best to ensure the paper's argument is clear, while also providing all the technical details to support each point in the Appendix. We have however incorporated the reviewer's specific suggestion in the revised manuscript, which now cites the experiment number in the relevant sections of both Results text and the figure legends.

*Reviewer #3:*

*In recent years the reliability of effects previously reported for tDCS have been questioned. Horvath and others argued that tDCS are either non-existent or smaller than previously assumed. It is suggested that reliable estimates for tDCS effects can only be expected with moderate to large sample sizes. In light of these comments, the sample sizes used in this study seem quite small.*

*RI: As noted above, an assessment of the power/sensitivity would be helpful here. I am familiar with the Horvath paper (and the responses to it). Certainly it’s important to note that there are concerns with reliability of tDCS.*

As described above (response to reviewer 1 point 1), power and sensitivity analyses confirm the adequacy of our sample sizes. The validity of Horvath et al's methodology has been called into question (Antal et al., 2015 Brain Stimulation). Yes, tDCS effects are variable. Our manuscript addresses this issue experimentally (Figure 4). Experiments of this nature inform about the underlying physiological sources of variability that can help explain inter-individual variance in behavioral response to stimulation. We now comment on this in Discussion.

Revised text (Discussion, seventh paragraph):

“Although it is well-replicated that, on average, M1 a-tDCS increases motor cortex excitability in healthy volunteers, stimulation effects vary across individuals. […] Experiments of this nature inform about the underlying physiological sources of variability that can help explain behavioural variability in response to stimulation.”

*1) Post-hoc comparisons using α-adjustments for multiple tests are avoided; instead one-tailed planned comparisons are carried out. To me this seems only acceptable if:*

a) The predictions are convincingly justified.b) The planned comparisons are limited to a small set of pre-determined conditions and variables.

*RI: Per above, please review your selection and justification for the statistical analyses.*

We agree with the reviewer and confirm that both criteria are fulfilled here:

a) Our a priori prediction was that M1 a-tDCS during PA would *enhance consolidation* of prism adaptation after-effects. This prediction was justified by a body of evidence in humans and animals indicating that M1 is a key cortical site for the early consolidation of motor memories. We now provide an expanded account of this literature and our rationale in the revised Introduction.

b) This *single,* specific, directional prediction (Anodal>Sham) is the only planned contrast we test (repeatedly in each experiment, where each experiment may consider a different outcome measure). Hence, a requirement for 'post hoc comparisons using α adjustments for multiple tests' does not arise. See also response to reviewer 2 point 7.

3) Neglect patients. All three neglect patients suffer both from neglect and hemianopia. Given that neglect and hemianopia are separate clinical entities associated with different types of brain damage and different etiologies this coincidence is quite surprising. Does this mean that we have a very unusual patient sample? Is it possible that severe neglect may have been misdiagnosed as hemianopia? There is hardly any information on the hemianopia and it seems the issue of hemianopia has been only addressed at the beginning. It would be interesting to learn whether the hemianopia also benefited from the treatment.

*RI: It would be useful to comment on the hemianopia issue and in particular, if your sample is atypical. If you have information on pre-post hemianopia testing, this could be interesting since I'm betting the hemianopia did not change (thus providing another argument for specificity of the improvement).*

Brain lesions, especially those caused by cerebrovascular accident, do not neatly respect nosological boundaries. Co-morbidities are common. Patients with neglect (especially when it is chronic and severe, as in our patients) more commonly present with a visual field deficit than without. Few studies aim to quantify this relative prevalence, but an exception is Cassidy et al., (1999), who report the combined prevalence to be three times more common than visual neglect alone. Thus, our patient sample is not unusual. In our study, visual field deficits are reported as part of the patients' clinical profile. They were not assessed as an outcome variable, as there is no evidence (and no rationale to suggest) that prism adaptation could change visual field defects. However, we can report anecdotally that Patient 3 went on subsequently to participate in a hemianopia rehabilitation trial. Hence, despite his neglect improving to ceiling levels on our test battery after prism/tDCS intervention (Figure 7—figure supplement 1), he still met criteria for entry into a clinical trial, indicating that his hemianopia persisted.

*4) To assess the clinical benefits neglect patients were repeatedly tested with the same paper and pencil tests. This is problematic. The test batteries do not provide a sufficient number of parallel test versions. Practice effects can be expected and may masquerade as treatment effects.*

*RI: You could acknowledge this limitation.*

We did use parallel test versions, and have now clarified this in the Materials and methods section.

Revised text (Materials and methods):

“To minimize general learning effects, parallel versions of tests were used for copy drawing, object cancellation, colouring and reading tests in both batteries.”

Our study design also aimed to further minimize non-specific practice effects by counter-balancing the order of tDCS interventions. Note that, across the four longitudinal patient timeseries, Patient 1 (phase A) and Patient 3 underwent stimulation in the order Anodal-Sham, while Patient 1 (phase B) and Patient 2 underwent stimulation in the order Sham-Anodal. Thus the (necessary) asymmetry in number of practice sessions preceding the 1^st^ and 2^nd^ prism therapy intervention (given a within-subjects design) was balanced across the statistical contrast of interest. It is therefore unlikely that the improvements in neglect observed in the anodal condition (but not in sham) merely reflect task practice. This point is stated clearly in the main text of the Results.

5) Neglect case series: AE retention is tested in patient 1 but not in patients 2-4. This is a shame it would be interesting to see whether AE retention correlates with the retention of clinical benefits. The group analysis on patients 2-4 seems problematic. I am not entirely sure but it seems that the analysis is carried out at the level of individual blocks and that blocks from one patient are mixed together with blocks from the other patients. I suspect that this procedure violates the assumption of statistical independence.

*RI: Per above in terms of considering your statistics.*

To test for correlation, a different design would be necessary (i.e. a large group patient study). Note that neither our data (nor the wider literature on prism therapy for neglect) necessarily imply that these two variables should correlate. Certainly, adaptation is required to cause both. However, regardless of stimulation, it is unclear whether there is a quantitative relationship between pointing and cognitive prism after-effects. Both could occur in parallel. One could gate the other, and stimulation could change the (non-linear) gain on that interaction. This question is for future work.

Re the group analysis, it is not the case that blocks from one patient are mixed together with blocks from the other patients. As stated previously, patient identity was modelled as a covariate. We have now added a sentence to the Methods to emphasize that this ensures intra- and inter-individual variance are not confounded.

Revised text (Methods, section “Statistical Analysis”):

“For all group analyses of patient data the factor 'Patient' was modeled as a covariate, to account for individual differences and to avoid confounding intra- and inter-individual variance.”

6) Naturalistic spatial reaching task. I could not find the results and analysis based on the findings from this task.

*RI: Please clarify.*

We apologise for this oversight. As mentioned above (Reviewer 2 point 9), this manuscript reports a large body of work (5 years of data). In our efforts to maintain narrative focus on the key points in the argument, previously we reported this control experiment only briefly. The rationale for this task was to test for any unintended negative side effects of our intervention on motor function, and, as stated previously, we did not observe any. However, we do recognise that this experiment should have been reported in full. We have now rectified this by providing full details in the Appendix and Table 1, together with a summary of findings from new statistical analysis in the main text Results and Discussion. The revised manuscript states that PA + M1 a-tDCS (versus sham) did not impair motor function on the kinematic task (as reported previously), but now reports further that patients' reach movement time was speeded in both hemifields.

Revised text (Results, section “Clinical Phase”):

“On each trial, the task required patients to perform a centre-out reaching movement with the ipsilesional (right) arm to grasp a ball, and then place it in a basket in the left or right hemifield, depending on a prior central visual instruction cue. […] Hence, the positive effects of stimulation-augmented prism therapy for visual neglect were accompanied by a motor facilitation effect on a task probe of normal everyday visually-guided reaching behaviour.”

Revised text (Table 1 legend):

“Table 1. Individual patients' reach kinematics before and after PA + tDCS. Patients 1 and 2 performed the task with the right arm once each in the days before and after PA was combined with M1 anodal or sham tDCS. […] Only the centre-out phase of the reach was analysed. PA combined with anodal tDCS (compared to sham) speeded centre-out reach movement time, regardless of whether the central pre-cue instructed a subsequent leftward or rightward movement after grasping the ball.”

Revised text (Discussion, second paragraph):

“The sustained leftward bias caused by PA + M1 a-tDCS that benefited visual neglect was not accompanied by (unintended) negative side effects for leftward motor behavior. A spatially cued reaching task (Table 1) revealed that these patients exhibited "directional hypokinesia"-like deficits, which are common in neglect (i.e. slower reach movement times when cued to the left compared to the right). This left-right asymmetry was modulated by PA + M1 a-tDCS, which speeded reach movement times overall. Hence, the benefits for visual neglect were accompanied by facilitation of motor performance.”

Revised text (Appendix, section “Naturalistic spatial reaching task”):

“Note therefore that the movement was identical on each trial, so the distinction between 'left' and 'right' trials was only that patients were cued in advance as to which of a leftward or rightward movement they would have to perform after they had grasped the ball. […] Statistical analyses asked whether PA and/or M1 a-tDCS would alter this expected pattern of behaviour.

**Revised text (Appendix):**

“9. Stimulation-augmented PA therapy for neglect facilitates reaching speed

Given that M1 stimulation induced a sustained leftward shift in reach accuracy without vision (ie: prism after-effect) and in spatial cognition (neglect), we tested for unintended negative motor side effects (i.e.: impaired leftward reaching). […]

In summary, the positive effects of PA + M1 a-tDCS for visual neglect were not accompanied by negative side effects on a motor task designed to measure everyday visually-guided reaching behaviour. In fact, stimulation combined with PA caused a generalized improvement in motor performance, reflected in faster reach movement times, regardless of whether a spatial pre-cue instructed subsequent movements to the left or to the right.

*7) Supplementary material: Section 7. For the ANOVA, effects from three sham sessions are combined and compared to the findings from one single anodal session. This seems problematic given that the sham sessions contained three times more trials than the anodal session.*

*RI: Per above in terms of considering your statistics.*

This is incorrect. As stated clearly in the text (pg. 55, relevant text underlined for emphasis), the RM ANOVA modelled the data separately for each of the 4 test sessions (i.e. 1 anodal, 3 sham). Hence it is not the case that the ANOVA combined effects from the three sham sessions such that the sham condition contained 3 times more trials than the anodal condition.

*8) Finally, a theoretical question: does the fact that M1-tDCS over the left hemisphere improved neglect signs really contradict the Kinsbourne/Corbetta hypothesis that hemispheric asymmetry is at the heart of the attentional deficit found in neglect patients? We know very little about the precise effects of tDCS on the brain – possibly not enough to exclude the possibility that left M1-tDCS may also lead to some reduction in left-hemispheric activation.*

RI: This is an interesting point to consider in terms of placing your work within current theoretical accounts of neglect.

We do not claim that our data "contradict the hypothesis that hemispheric asymmetry is at the heart of the attentional deficit in neglect". We do not see anywhere in our manuscript where this claim is made. Our study does not address this question. There is a well-established body of evidence supporting a causal role for left hemisphere over-excitation in the pathophysiology of neglect, and neglect improves when this is suppressed. Our manuscript details these facts in the Discussion. We also state explicitly both at the start of our paper (Abstract) and at the very end (final paragraph of Discussion) that: "the left hemisphere in neglect is pathologically over-excited". Hence, our paper in no way contradicts the hypothesis that hemispheric asymmetry is at the heart of the attentional deficit in neglect. To avoid this misunderstanding, we now state this explicitly.

Rather, our paper challenges the consensus in the literature that becauseof this left hemisphere over-excitation, to improve neglect, the left hemisphere should be suppressed. While this strategy is known to be effective, our data show that suppression is not necessaryto improve neglect. Instead, we present a conceptually novel approach, supported by evidence, that intervening to excite the left hemisphere (to augment its adaptive capacity) can produce lasting improvements.